# *Cortex cis*-regulatory switches establish scale colour identity and pattern diversity in *Heliconius*

**Luca Livraghi**[1,2]*, **Joseph J Hanly**[1,2,3], **Steven M Van Bellghem**[4],
**Gabriela Montejo-Kovacevich**[1], **Eva SM van der Heijden**[1,2], **Ling Sheng Loh**[3],
**Anna Ren**[3], **Ian A Warren**[1], **James J Lewis**[5], **Carolina Concha**[2],
**Laura Hebberecht**[1,2], **Charlotte J Wright**[1], **Jonah M Walker**[1], **Jessica Foley**[2],
**Zachary H Goldberg**[6], **Henry Arenas-Castro**[2], **Camilo Salazar**[7], **Michael W Perry**[6],
**Riccardo Papa**[4], **Arnaud Martin**[3], **W Owen McMillan**[2], **Chris D Jiggins**[1,2]

[1]Department of Zoology, University of Cambridge, Downing St., Cambridge, United Kingdom; [2]Smithsonian Tropical Research Institute, Gamboa, Panama; [3]The George Washington University Department of Biological Sciences, Science and Engineering Hall, Washington, United States; [4]Department of Biology, Centre for Applied Tropical Ecology and Conservation, University of Puerto Rico, Rio Piedras, Puerto Rico; [5]Baker Institute for Animal Health, College of Veterinary Medicine, Cornell University, Ithaca, United States; [6]Cell & Developmental Biology, Division of Biological Sciences, UC San Diego, La Jolla, United States; [7]Biology Program, Faculty of Natural Sciences, Universidad del Rosario, Bogotá, Colombia

**\*For correspondence:**
miles.livraghi@gmail.com

**Competing interests:** The authors declare that no competing interests exist.

**Abstract** In *Heliconius* butterflies, wing colour pattern diversity and scale types are controlled by a few genes of large effect that regulate colour pattern switches between morphs and species across a large mimetic radiation. One of these genes, *cortex*, has been repeatedly associated with colour pattern evolution in butterflies. Here we carried out CRISPR knockouts in multiple *Heliconius* species and show that *cortex* is a major determinant of scale cell identity. Chromatin accessibility profiling and introgression scans identified *cis*-regulatory regions associated with discrete phenotypic switches. CRISPR perturbation of these regions in black hindwing genotypes recreated a yellow bar, revealing their spatially limited activity. In the *H. melpomene/timareta* lineage, the candidate CRE from yellow-barred phenotype morphs is interrupted by a transposable element, suggesting that *cis*-regulatory structural variation underlies these mimetic adaptations. Our work shows that *cortex* functionally controls scale colour fate and that its *cis*-regulatory regions control a phenotypic switch in a modular and pattern-specific fashion.

## Introduction

Butterfly wing pattern diversity provides a window into the ways genetic changes underlie phenotypic variation that is spatially limited to specific parts or regions of the organism (*McMillan et al., 2020*; *Orteu and Jiggins, 2020*; *Rebeiz et al., 2015*). Many of the underlying genetic loci controlling differences in colour patterns have been mapped to homologus 'hotspots' across disparate taxa. In some cases, this repeated adaptation has occurred through the alteration of downstream effector genes, such as pigment biosynthetic enzymes with functions clearly related to the trait under selection, for example, the genes *tan* and *ebony* that control insect melanin pigmentation (reviewed in *Massey and Wittkopp, 2016*). In other cases, upstream regulatory genes are important, and these are typically either transcription factors (e.g. *optix, MITF, Sox10*) or components of signalling

**eLife digest** *Heliconius* butterflies have bright patterns on their wings that tell potential predators that they are toxic. As a result, predators learn to avoid eating them. Over time, unrelated species of butterflies have evolved similar patterns to avoid predation through a process known as Müllerian mimicry. Worldwide, there are over 180,000 species of butterflies and moths, most of which have different wing patterns. How do genes create this pattern diversity? And do butterflies use similar genes to create similar wing patterns?

One of the genes involved in creating wing patterns is called *cortex*. This gene has a large region of DNA around it that does not code for proteins, but instead, controls whether *cortex* is on or off in different parts of the wing. Changes in this non-coding region can act like switches, turning regions of the wing into different colours and creating complex patterns, but it is unclear how these switches have evolved.

Butterfly wings get their colour from tiny structures called scales, which each have their own unique set of pigments. In *Heliconius* butterflies, there are three types of scales: yellow/white scales, black scales, and red/orange/brown scales. Livraghi et al. used a DNA editing technique called CRISPR to find out whether the *cortex* gene affects scale type.

First, Livraghi et al. confirmed that deleting *cortex* turned black and red scales yellow. Next, they used the same technique to manipulate the non-coding DNA around the *cortex* gene to see the effect on the wing pattern. This manipulation turned a black-winged butterfly into a butterfly with a yellow wing band, a pattern that occurs naturally in *Heliconius* butterflies. The next step was to find the mutation responsible for the appearance of yellow wing bands in nature. It turns out that a bit of extra genetic code, derived from so-called 'jumping genes', had inserted itself into the non-coding DNA around the *cortex* gene, 'flipping' the switch and leading to the appearance of the yellow scales.

Genetic information contains the instructions to generate shape and form in most organisms. These instructions evolve over millions of years, creating everything from bacteria to blue whales. Butterfly wings are visual evidence of evolution, but the way their genes create new patterns isn't specific to butterflies. Understanding wing patterns can help researchers to learn how genetic switches control diversity across other species too.

pathways such as ligands or receptors (e.g. *WntA*, *MC1R*, *Agouti*). These 'developmental toolkit genes' influence pigment cell fate decisions by modulating gene regulatory networks (GRNs) (*Kronforst and Papa, 2015*; *Martin and Courtier-Orgogozo, 2017*; *Prud'homme et al., 2007*), and are commonly characterised by highly conserved functions, with rapid evolutionary change occurring through regulatory fine-tuning of expression patterns. One gene that has been repeatedly implicated in morphological evolution but does not conform to this paradigm is *cortex*, a gene implicated by mapping approaches in the regulation of adaptive changes in the wing patterning of butterflies and moths.

*Cortex* is one of four major effect genes that act as switch loci controlling both scale structure and colour patterns in *Heliconius* butterflies, and has been repeatedly targeted by natural selection to drive differences in pigmentation (*Nadeau, 2016*; *Van Belleghem et al., 2017*). Three of the four major effect genes correspond to the prevailing paradigm of highly conserved patterning genes; the signalling ligand *WntA* (*Martin et al., 2012*; *Mazo-Vargas et al., 2017*) and two transcription factors *optix* (*Lewis et al., 2019*; *Reed et al., 2011*; *Zhang et al., 2017*) and *aristaless1* (*Westerman et al., 2018*). *Cortex*, on the other hand, is an insect-specific gene showing closest homology to the *cdc20/fizzy* family of cell cycle regulators (*Chu et al., 2001*; *Nadeau et al., 2016*; *Pesin and Orr-Weaver, 2007*). The lepidopteran orthologue of *cortex* displays rapid sequence evolution and has acquired novel expression domains that correlate with melanic wing patterns in *Heliconius* (*Nadeau et al., 2016*; *Saenko et al., 2019*). It therefore seems likely that the role of *cortex* in regulating wing patterns has involved a major shift in function, which sits in contrast to the classic model of regulatory co-option of deeply conserved patterning genes.

The genetic locus containing *cortex* was originally identified in the genus *Heliconius* as controlling differences in yellow and white wing patterns in *H. melpomene* and *H. erato* (*Figure 1a*) and the

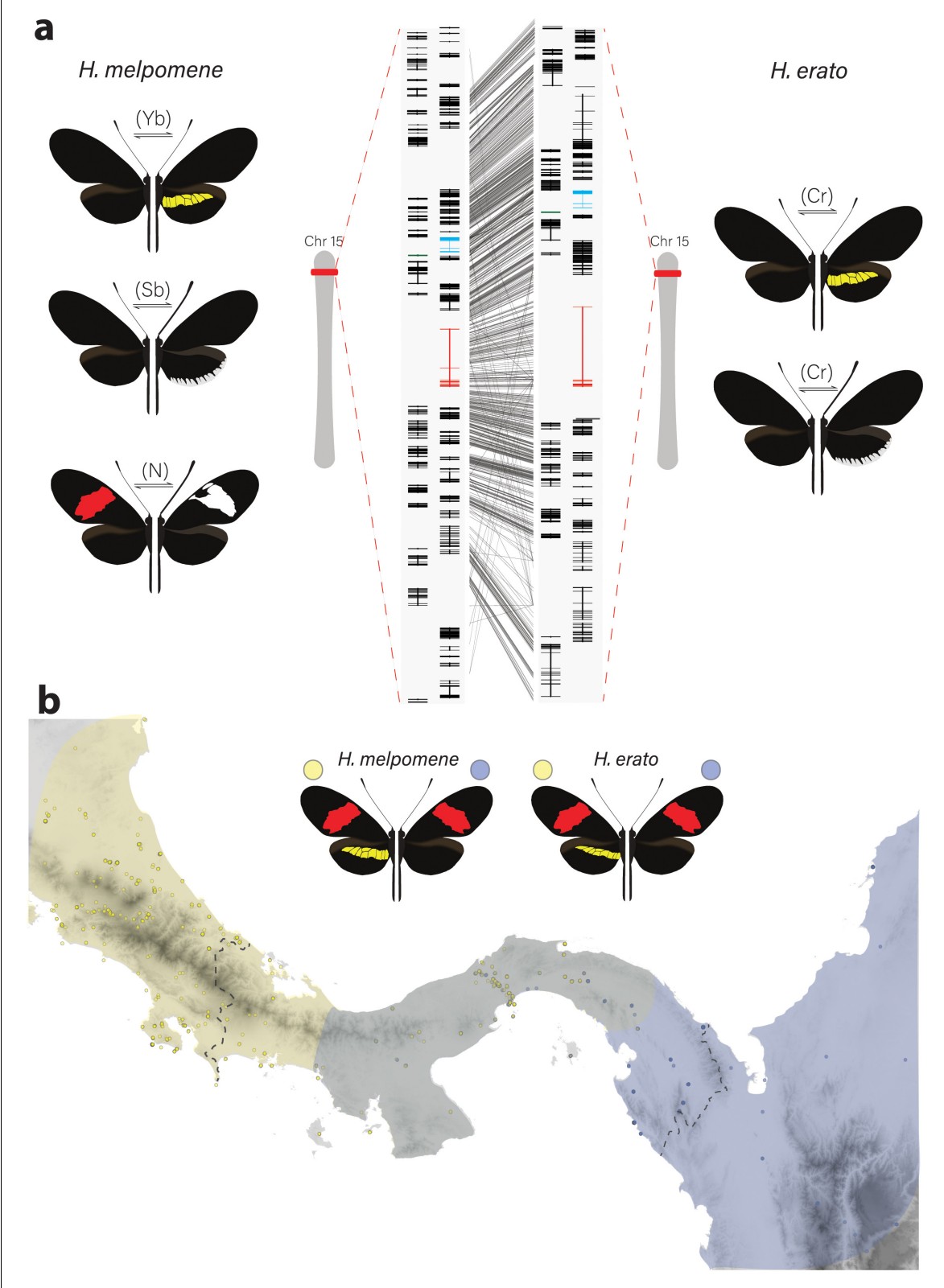

**Figure 1.** Phenotypic switches of yellow and white colour pattern elements are controlled by homologous loci in *Heliconius* species. (a) Homologous loci in both *H. erato* and *H. melpomene* are associated with variation in yellow and white patterns between morphs. In *H. melpomene*, three tightly linked genetic elements located at chromosome 15 have been identified that control variation for the hindwing yellow bar, white margin elements, and forewing band (Yb, Sb, and N, respectively), while in *H. erato*, variation has been mapped to one element (Cr). The gene *cortex* is highlighted in red,

*Figure 1 continued on next page*

*Figure 1 continued*

*dome/wash* in blue, and *dome-trunc* in green. Alignments between the two co-mimetic species at the locus is shown (grey lines, 75% alignment identity). Gene models from assemblies of *H. melpomene* (left) and *H. erato* (right) are shown for the loci spanning the associated intervals controlling the phenotypic switches highlighted. Horizontal bars indicate exons, vertical bars introns. (**b**) Focal co-mimetic morphs of *H. erato* and *H. melpomene* used in this study, differing in the presence of a hindwing yellow bar, and their ranges across Central America are shown. Yellow: yellow banded morphs, blue: black hindwing morphs, grey: range overlap. Each dot represents a sampled location (data from *Rosser et al., 2012*). Country borders are indicated by dotted lines.

The online version of this article includes the following figure supplement(s) for figure 1:

**Figure supplement 1.** Maximum-likelihood tree based on lepidopteran *dome* amino acid sequences.

**Figure supplement 2.** Annotation of the genes present in the 47 gene interval previously shown to be associated with colour pattern differences in *Heliconius*.

**Figure supplement 3.** To examine the conservation of *dome/wash* bi-cistronic transcription in the Lepidoptera, we performed BLASTn searches using the previously annotated *dome* transcripts from the *H*.

**Figure supplement 4.** ATAC-sequencing analysis supports *dome/wash* bi-cistronic transcription in *Heliconius erato*.

polymorphism in yellow, white, black, and orange elements in *H. numata*. This was inferred using a combination of association mapping and gene expression data (*Joron et al., 2006*; *Nadeau et al., 2016*). The same locus has also been repeatedly implicated in controlling colour pattern variation among divergent Lepidoptera, including the peppered moth *Biston betularia* and other geometrids, the silkmoth *Bombyx mori*, and other butterflies such as *Junonia coenia*, *Bicyclus anynana*, and *Papilio clytia* (*Beldade et al., 2009*; *van der Burg et al., 2020*; *Ito et al., 2016*; *VanKuren et al., 2019*; *Van't Hof et al., 2019*; *Van't Hof et al., 2016*). This locus therefore contains one or more genes that have repeatedly been targeted throughout the evolutionary history of the Lepidoptera to generate phenotypic diversity.

In *Heliconius* butterflies, population genomic data suggest that *cis*-regulatory modules surrounding *cortex* underlie adaptive variation of yellow and white colour pattern elements (*Enciso-Romero et al., 2017*; *Van Belleghem et al., 2017*). These studies predict the existence of modular elements that compartmentalise expression of colour pattern genes across developing wings. However, developmental genes have complex regulatory domains and recent work has suggested that pleiotropy among different enhancers may be more common than is currently appreciated (*Lewis et al., 2019*; *Murugesan et al., 2021*; *Nagy et al., 2018*). Further dissection of the regulatory elements controlling wing pattern variation is thus necessary to assess the relative contribution of pleiotropy versus modularity at colour pattern loci (*Lewis and Van Belleghem, 2020*).

While fantastically diverse, most of the pattern variation in *Heliconius* is created by differences in the distribution of just three major scale cell types: Type I (yellow/white), Type II (black), and Type III (red/orange/brown) (*Aymone et al., 2013*; *Gilbert et al., 1987*). Each type has a characteristic nanostructure and a fixed complement of pigments. Type I yellow scales contain the ommochrome precursor 3-hydroxykynurenine (3-OHK) (*Finkbeiner et al., 2017*; *Koch, 1993*; *Reed et al., 2008*), whereas Type I white scales lack pigment, and the white colour is the result of the scale cell ultrastructure (i.e. structural white) (*Gilbert et al., 1987*) (see Figure 9f). Structurally, Type I scales are characterised by the presence of a lamina covering the scale windows and by microribs joining the larger longitudinal ridges. In contrast, Type II scale cells are pigmented with melanin, have larger crossribs and lack a lamina covering the scale windows. Quantitative variation in scale structures between populations (but not within individuals) can cause Type II scales to range from matte black to iridescent blue (*Brien et al., 2019*; *Parnell et al., 2018*). Finally, Type III scale cells contain the red ommochrome pigments xanthommatin and dihydroxanthommatin and are characterised by larger spacing between crossribs and ridges.

Here we focus on the role of *cortex* in specifying these scale types in *Heliconius* butterflies, an adaptive radiation with over 400 different wing forms in 48 described species (*Jiggins, 2017*; *Lamas, 2004*) and where diversity in wing patterns can be directly linked to the selective forces of predation and sexual selection (*Brown, 1981*; *Turner, 1981*). Specifically, we combine expression profiling using *RNA-seq*, *ATAC-seq*, in situ hybridisation and antibody staining experiments, as well as CRISPR/Cas9 gene knockouts to determine the role that this locus plays in pattern variation of two co-mimetic morphs of *H. melpomene* and *H. erato* (*Figure 1b*). We focus on two mimetic morphs differing specifically in the presence/absence of a yellow hindwing bar, whose phenotypic

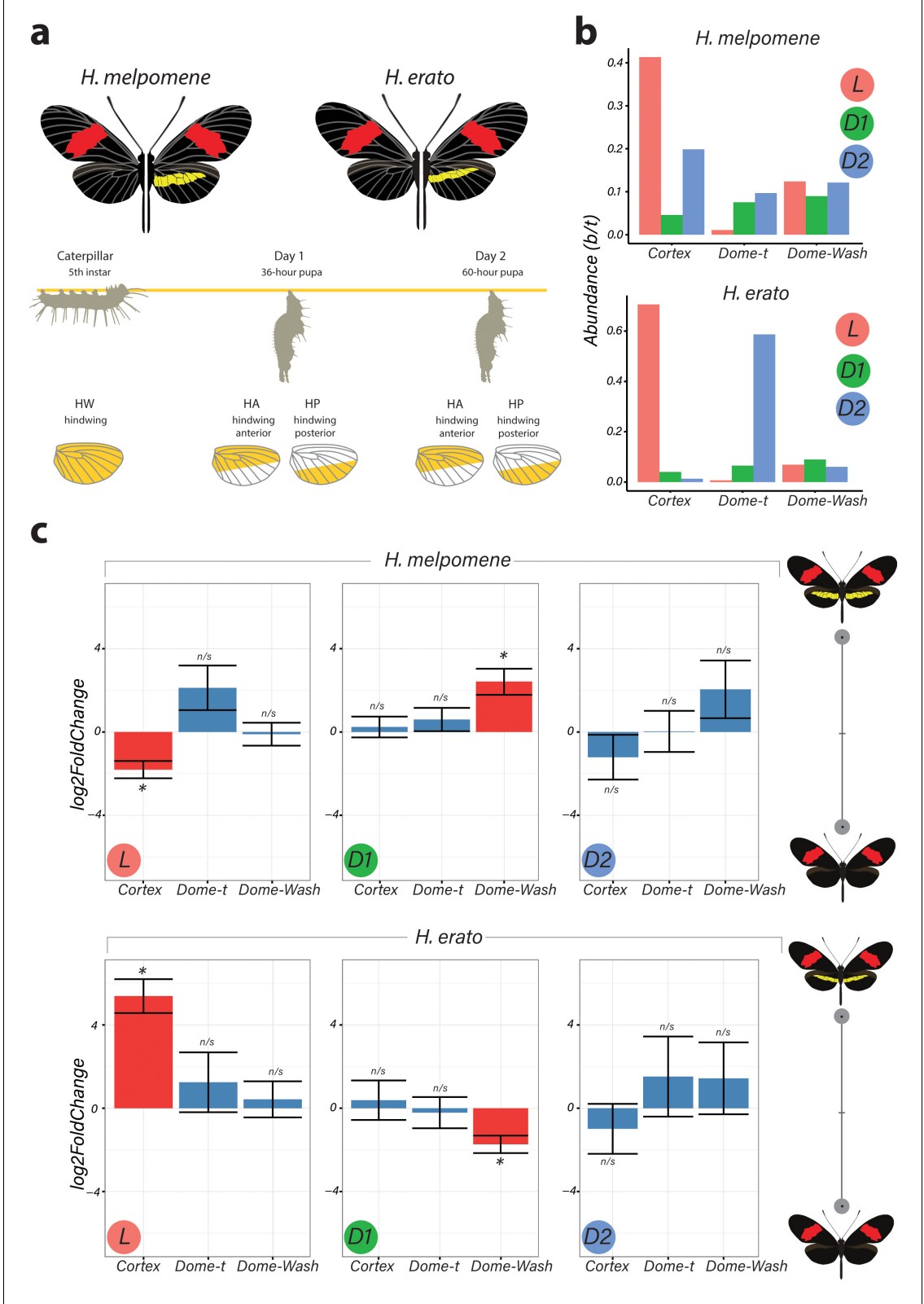

**Figure 2.** Differential expression of genes at Chromosome 15 implicate *cortex* as most likely candidate driving yellow bar differences. (a) Hindwing tissue from co-mimetic morphs of *H. melpomene* and *H. erato* were collected at three developmental stages fifth-instar caterpillar, Day 1 Pupae (36hAPF) and Day 2 Pupae (60hAPF). For pupal tissue, hindwing tissue was dissected using the wing vein landmarks shown, corresponding to the future adult position of the hindwing yellow bar (dissection scheme based on **Hanly et al., 2019**). (b) Relative abundance of transcripts corresponding to the *Figure 2 continued on next page*

*Figure 2 continued*

genes *cortex, domeless-truncated, domeless/washout* throughout developmental stages. (c) Log$_2$FoldChange for the genes *cortex, domeless-truncated (dome-t), domeless/washout (dome-wash)* across developmental stages. Comparisons are for whole wing discs (Larvae, L) and across wing sections differing in the presence of a yellow bar in pupal wings (D1 and D2; see *Figure 2—figure supplement 2*: for depiction of contrasts analysed). *Adjusted p<0.05; n/s = not significant. N = 3 for each bar plot.

The online version of this article includes the following source data and figure supplement(s) for figure 2:

**Source data 1.** RNA-seq samples were genotyped relative to protein-coding WGS SNPs from individuals from the source populations in Panama.
**Source data 2.** This was repeated for the *H. erato* samples; here, only one informative protein-coding SNP was found, in the gene *parn*.
**Source data 3.** Primers used for qPCR experiments for housekeeping genes and *cortex* are shown below.
**Source data 4.** Gene IDs in the *H. melpomene* Yb locus and their corresponding IDs in the *H. erato* genome.
**Figure supplement 1.** qPCR confirms direction of differential expression in Heliconius erato.
**Figure supplement 2.** DGE analysis shows *cortex* and *dome/wash* are consistently differentially expressed between colour pattern races and pupal wing sections.
**Figure supplement 3.** Differential expression across the *cortex* locus in *H. melpomene*, shown as the negative log of the adjusted p-value (–log(padj)).
**Figure supplement 4.** Depiction of contrasts.
**Figure supplement 5.** analysis of the cdc20/cdh1 family reveals Cortex is a derived and insect-specific derivative of cdc20.
**Figure supplement 6.** Full Cdc20 family protein alignments (see legend in *Figure 2—figure supplement 5* for abbreviations).

switch has been mapped to non-coding regions surrounding the gene *cortex*. We also test its function in diverse patterning morphs, including ones differing in the presence of white margin elements spanning the hindwing, as well as species displaying divergent and complex phenotypes such as the tiger striped silvaniform *Heiconius hecale*. Despite *cortex* not following the prevailing paradigm of patterning loci, we demonstrate that the gene plays a fundamental role in pattern variation by modulating a switch from Type I scale cells to Type II and Type III scale cells. Moreover, we show that while the phenotypic effects of *cortex* extend across the entire fore- and hindwing surface, modular enhancers have evolved in two distantly related *Heliconius* species that control spatially restricted, pattern-specific expression of *cortex*. Our findings, coupled with recent functional experiments on other *Heliconius* patterning loci, are beginning to illuminate how major patterning genes interact during development to determine scale cell fate and drive phenotypic variation across a remarkable adaptive radiation.

## Results

### The genes *cortex* and *domeless/washout* are differentially expressed between colour pattern morphs and between wing sections differing in the presence of the hindwing yellow bar

To identify genes associated with the yellow bar phenotype, we performed differential gene expression (DGE) analysis using developing hindwings sampled from colour pattern morphs in *H. erato* and *H. melpomene* differing only in the presence or absence of the hindwing yellow bar (*Figures 1b* and *2a*). In total, we sequenced 18 samples representing three developmental stages (larval, 36 h ± 1.5 h [Day 1 pupae] and 60 h ± 1.5 h [Day 2 pupae]) from two morphs in each of the two species, with hindwings divided into two parts for the pupal stages (*Figure 2a*). We focused our attention on genes centred on a 47-gene interval on chromosome 15 previously identified as the minimal associated region with yellow band phenotypes by recombination and population genetic association mapping (*Nadeau et al., 2016*, Supp Table 1; *Joron et al., 2006*; *Moest et al., 2020*; *Van Belleghem et al., 2017*). Both our initial expression analysis and recent analysis of selective sweeps at this locus (*Moest et al., 2020*) indicate that three genes show differential expression and are likely targets of selection: *cortex, domeless (dome)*, and *washout (wash)* (*Figure 2c*). In *Heliconius, dome* appears to have duplicated in the ancestor of *H. erato* and *H. melpomene*, resulting in a full-length copy (referred to here as *domeless*) and a further copy exhibiting truncations at the C-terminus (*domeless-truncated [dome-trunc]*) (*Figure 1—figure supplements 1–2*). Transcriptomic and previous evidence (*Lewis et al., 2020*) also indicates that *dome* and *wash* are transcribed as a single bi-cistronic gene (*Figure 1—figure supplements 3–4*). Differential expression analysis was thus performed with *dome/wash* as a single annotation.

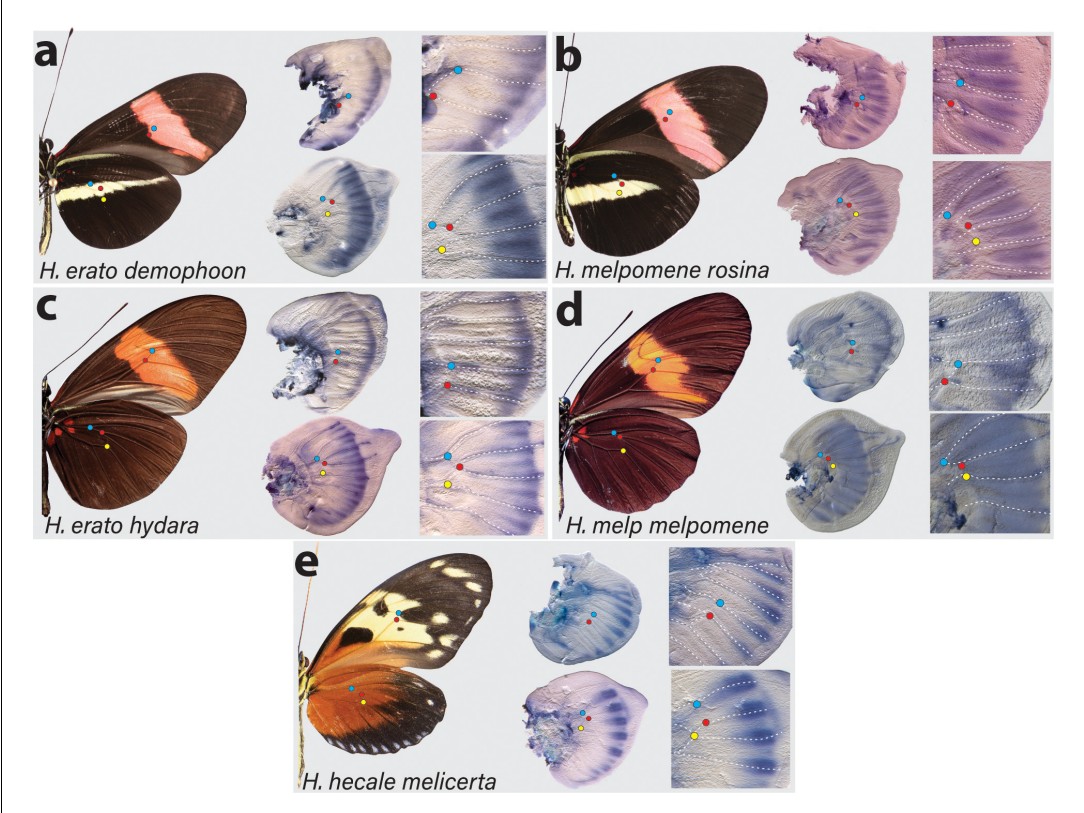

**Figure 3.** Expression of *cortex* transcripts in *Heliconius melpomene*, *Heliconius erato*, and *Heliconius hecale* fifth-instar wing discs. *Cortex* expression in fifth-instar wing discs is restricted to the distal end of both forewings and hindwings in all species and morphs analysed. In *H. erato*, expression is strongest at the intervein midline but extends across vein compartments in *H. erato demophoon* (a), whereas it is more strongly localised to the intervein midline in *H. erato hydara* (c). In *H. melpomene rosina* (b), *cortex* localises in a similar manner to *H. erato demophoon*, with stronger expression again observed at the intervein midline, whereas expression in *H. melpomene melpomene* (d) extends more proximally. Expression in *H. hecale melicerta* (e) is strongest at the distal wing vein margins. Coloured dots represent homologous vein landmarks across the wings.

The online version of this article includes the following figure supplement(s) for figure 3:

**Figure supplement 1.** Distal expression of *cortex* in *Heliconius* fifth-instar imaginal discs.

The two species were analysed separately, with both showing only *cortex* and *dome/wash* as significantly differentially expressed between morphs among the 47 genes in the candidate region, with *cortex* differential expression occurring earlier in development. In fifth-instar larvae, *cortex* is differentially expressed in both species between the two colour pattern morphs, with *cortex* showing the highest adjusted p-value (Benjamini and Hochberg correction) for any gene in the genome at this stage in *H. erato* (*Figure 2c*). Interestingly, *cortex* transcripts were differentially expressed in opposite directions in the two species, with higher expression in the melanic hindwing morph in *H. melpomene* and in the yellow banded morph in *H. erato*. We confirmed this pattern of expression through a SNP association analysis (*Figure 2—source data 1–3*) and RT-qPCR (*Figure 2—figure supplement 1*). This pattern is reversed for *dome/wash* in Day 1 pupae, where a statistically higher proportion of transcripts are detected in *H. melpomene rosina* (yellow) and in *H. erato hydara* (melanic) (*Figure 2—figure supplements 2–4*). No differential expression of these genes was found at Day 2 pupae.

When comparing across hindwing sections differing for the yellow bar phenotype, 22 genes of the associated 47-gene interval were differentially expressed at Day 1 between relevant wing sections in *H. melpomene*, including *cortex* and *dome/wash* (*Figure 2—figure supplements 2–4*). In contrast, in *H. erato* Day 1 pupae, only *dome/wash* was differentially expressed. At Day 2 pupae, there were no differentially expressed genes in either species between relevant wing sections at this locus.

Given the strong support for the involvement of *cortex* in driving wing patterning differences, we re-analysed its phylogenetic relationship to other cdc20 family genes with more extensive sampling than previous analyses (*Nadeau et al., 2016*). Our analysis finds strong monophyletic support for *cortex* as an insect-specific member of the cdc20 family, with no clear *cortex* homologs found outside of the Neoptera (*Figure 2—figure supplement 5*). Branch lengths indicate *cortex* is evolving rapidly within the lineage, despite displaying conserved anaphase promoting complex (APC/C binding motifs, including the C-Box and IR tail *Figure 2—figure supplement 6*; *Chu et al., 2001*; *Pesin and Orr-Weaver, 2007*).

In summary, *cortex* is the most consistently differentially expressed gene and showed differential expression earlier in development as compared to the other candidate *dome/wash*. We therefore focused subsequent experiments on *cortex*, although at this stage we cannot rule out an additional role for *dome/wash* in yellow pattern specification.

## *Cortex* transcripts localise distally in fifth-instar larval wing discs

Two studies have reported that *cortex* mRNA expression correlates with melanic patches in two species of *Heliconius* (*Nadeau et al., 2016*; *Saenko et al., 2019*). To further assess this relationship between *cortex* expression and adult wing patterns, we performed in situ hybridisation on developing wing discs of fifth-instar larvae, where we observed largest *cortex* transcript abundance, in both the yellow-barred and plain hindwing morphs of *H. erato* and *H. melpomene*. *Cortex* transcripts at this stage localised distally in forewings and hindwings of both species (*Figure 3—figure supplement 1*). In *H. erato demophoon* hindwings, expression was strongest at the intervein midline, but extends across vein compartments covering the distal portion of both forewing and hindwing (*Figure 3a*). By contrast, in *H. erato hydara* hindwings, *cortex* transcripts are more strongly localised to the intervein midline forming a sharper intervein expression domain (*Figure 3c*).

Expression in *H. melpomene rosina* is similar to *H. erato demophoon* at comparable developmental stages, again with stronger expression localised to the intervein midline but extending further proximally than in *H. erato demophoon* (*Figure 3b*). In *H. melpomene melpomene*, hindwing *cortex* expression extends across most of the hindwing, and does not appear to be restricted to the intervein midline (*Figure 3c*).

Given that *cortex* has been implicated in modulating wing patterns in many divergent lepidoptera, we also examined localisation in a *Heliconius* species displaying distinct patterns: *H. hecale melicerta* (*Figure 3e*). Interestingly, in this species, transcripts appear strongest in regions straddling the wing disc veins, with weak intervein expression observed only in the hindwings. Previous data has shown that variation in yellow spots (Hspot) is also controlled by a locus located a chromosome 15 (*Huber et al., 2015*). Expression in *H. hecale melicerta* forewings corresponds to melanic regions located in between yellow spots at the wing margins, indicating *cortex* may be modulating Hspot variation in *H. hecale*.

Overall, our results suggest a less clear correlation to melanic elements than reported expression patterns (*Nadeau et al., 2016*; *Saenko et al., 2019*) where *cortex* expression in fifth-instar caterpillars is mostly restricted to the distal regions of developing wings, but appears likely to be dynamic across fifth-instar development (*Figure 3—figure supplement 1*).

## *Cortex* establishes type II and III scale identity in *Heliconius* butterflies

To assay the function of *cortex* during wing development, we generated $G_0$ somatic mosaic mutants using CRISPR/Cas9 knock outs. We targeted multiple exons using a combination of different guides and genotyped the resulting mutants through PCR amplification, cloning, and Sanger sequencing (*Figure 4—figure supplement 1*). Overall KO efficiency was low when compared to similar studies in *Heliconius* (*Concha et al., 2019*; *Mazo-Vargas et al., 2017*), with observed wing phenotype to hatched eggs ratios ranging from 0.3% to 4.8%. Lethality was also high, with hatched to adult ratios ranging from 8.1% to 29.8% (*Figure 4—source data 1–2*).

Targeting of the *cortex* gene in *H. erato* morphs produced patches of ectopic yellow and white scales spanning regions across both forewings and hindwings (*Figure 4—figure supplements 2–4*). All colour pattern morphs were affected in a similar manner in *H. erato*. Mutant clones were not restricted to any specific wing region, affecting scales in both proximal and distal portions of wings. The same effect on scale pigmentation was also observed in *H. melpomene* morphs, with mutant

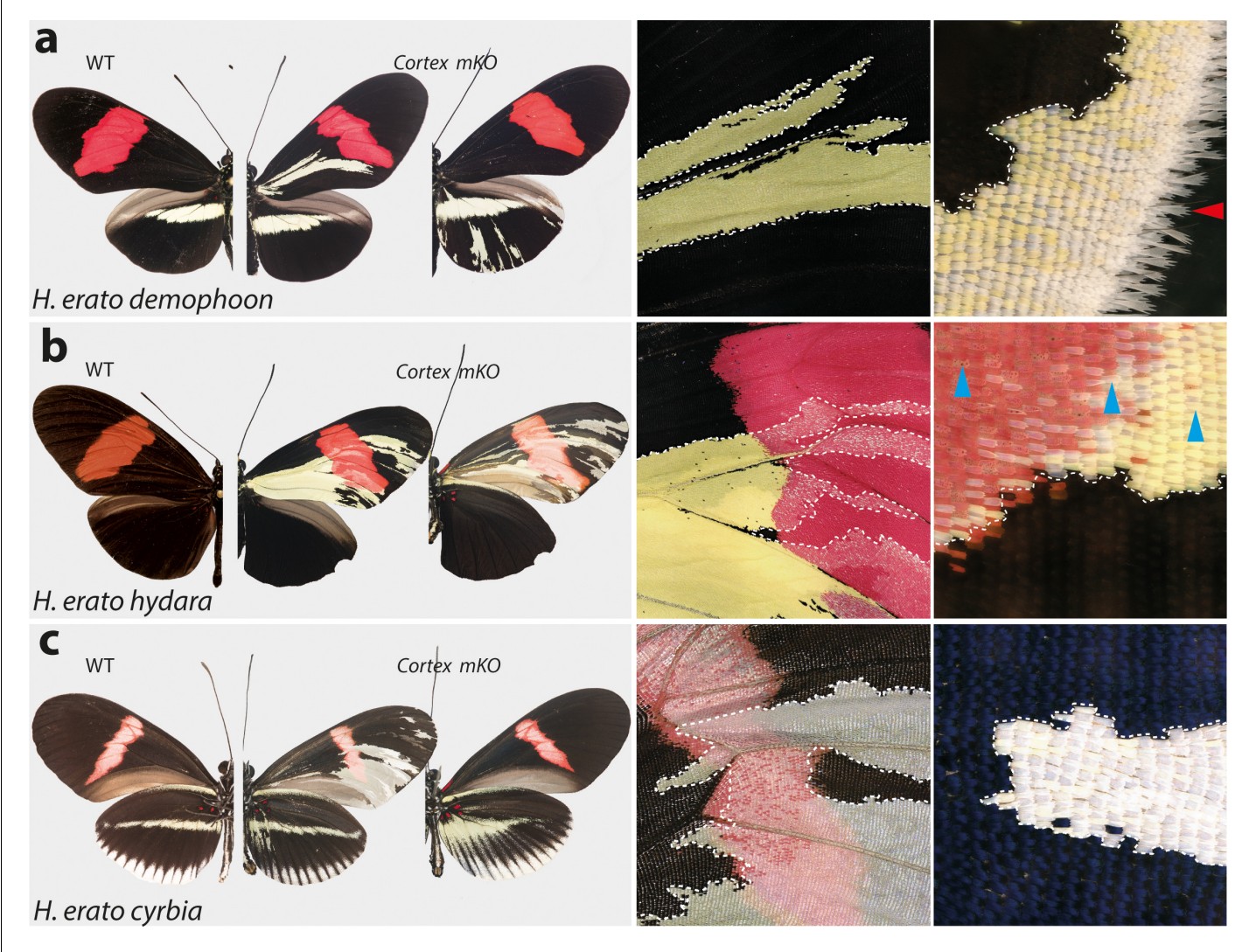

**Figure 4.** *Cortex* loss-of-function transforms scale identity across the entire wing surface of *Heliconius erato*. Phenotypes of *cortex* mKO across *H. erato* morphs reveal a loss of melanic (Type II) and red (Type III) scales and transformation to Type I (yellow or white) scales. Affected regions are not spatially restricted and span both distal and proximal portions of forewings and hindwings. The scale transformation extends to all scale types, including the wing border scales (red arrow head in (**a**)), and across the red band elements, where mutant scales transform to white, as well as some showing an intermediate phenotype (blue arrow heads in (**b**)). A positional effect is observed in some morphs, where ectopic Type I scales are either white or yellow depending on their position along the wing (*H. erato cyrbia*, (**c**)). Ectopic Type I scales can be induced from both melanic and red scales, switching to either white or yellow depending on wing position and morph. Boundaries between Wild-type (WT) to mutant scales are highlighted (dotted white line).

The online version of this article includes the following source data and figure supplement(s) for figure 4:

**Source data 1.** CRISPR experiments and guides used per species/morph.

**Source data 2.** Sequences for guides yielding successful phenotypes and associated genotyping primers.

**Figure supplement 1.** CRISPR mutagenesis confirmed through sanger sequencing.

**Figure supplement 2.** *H. erato cyrbia* wild-type (WT), alongside *cortex* mKO individuals recovered in CRISPR experiments.

**Figure supplement 3.** *H. erato demophoon* wild-type (WT), alongside *cortex* mKO individuals recovered in CRISPR experiments.

**Figure supplement 4.** *H. erato hydara* wild-type (WT), alongside *cortex* mKO individuals recovered in CRISPR experiments.

**Figure supplement 5.** Mutant close-ups illustrating variety of effects caused by *cortex* mKO.

clones affecting both distal and proximal regions in forewings and hindwings (*Figure 5a–c*). In *H. erato hydara*, we recovered mutant individuals where clones also spanned the red forewing band (*Figure 4b*, *Figure 4—figure supplement 5*). Clones affecting this region caused what appears to

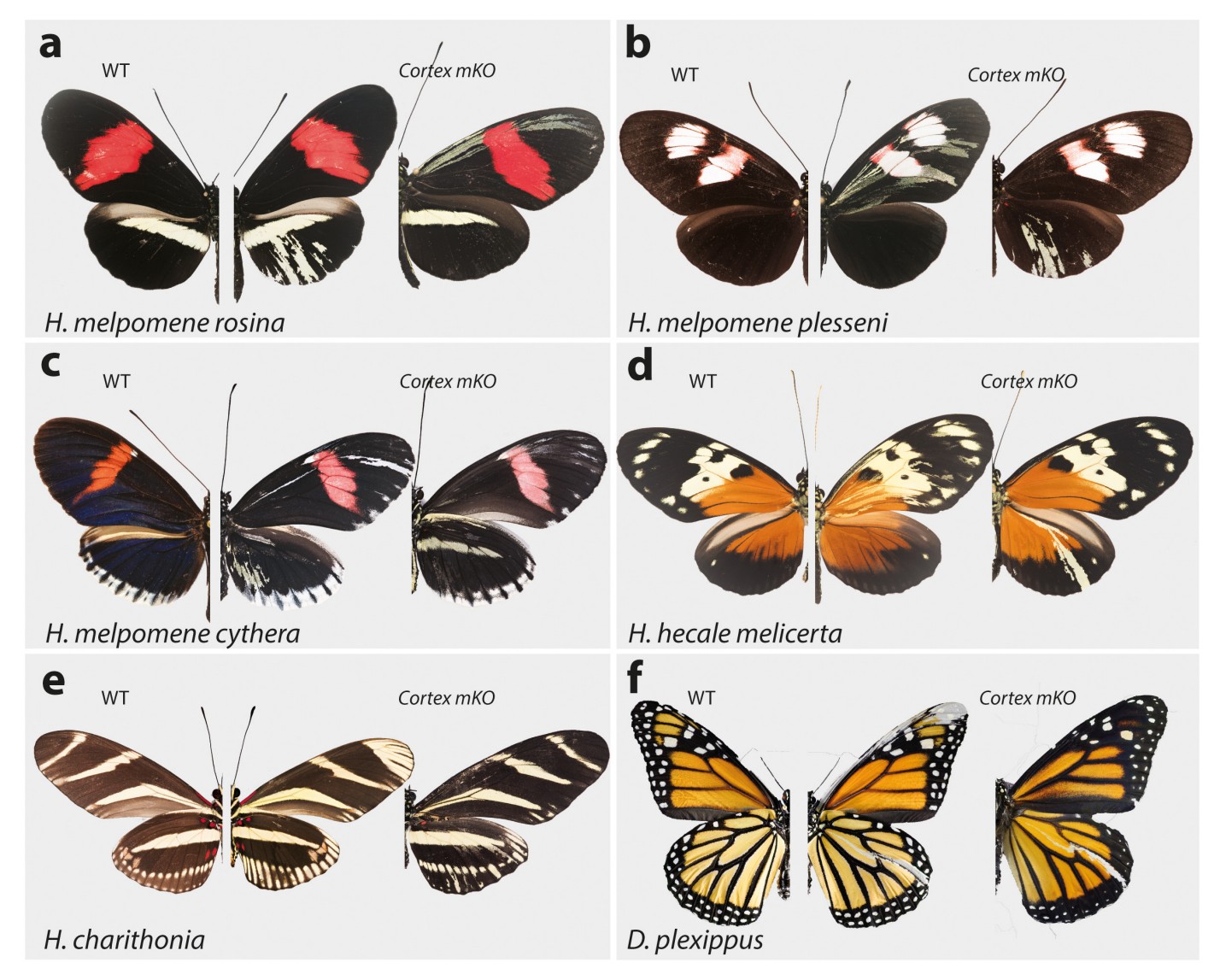

**Figure 5.** *Cortex* function is conserved across *Heliconius* and Nymphalids Phenotypes of *cortex* mKO across *H. melpomene* colour pattern morphs. (**a–c**) reveal *cortex* has a conserved function in switching scale cell fates, as in *H. erato*. This function is also conserved in outgroups to *H. melpomene* and *H. erato* (*H. hecale melicerta* and *H. charithonia* respectively (**d–e**)) as well as in distantly diverged nymphalids (*D. plexippus* (**f**)). Left; wild-type, middle and right; *cortex* mKO.

The online version of this article includes the following figure supplement(s) for figure 5:

**Figure supplement 1.** *H. melpomene plesseni* wild-type (WT), alongside *cortex* mKO individuals recovered in CRISPR experiments.

**Figure supplement 2.** *H. melpomene cythera* wild-type (WT), alongside *cortex* mKO individuals recovered in CRISPR experiments.

**Figure supplement 3.** *H. hecale melicerta* wild-type (WT), alongside *cortex* mKO individuals recovered in CRISPR experiments.

**Figure supplement 4.** *H. charithonia* wild-type (WT), alongside *cortex* mKO individuals recovered in CRISPR experiments.

**Figure supplement 5.** *Danaus plexippus* wild-type (WT), alongside *cortex* CRE mKO individuals recovered in CRISPR experiments.

be an asymmetric deposition of pigment across the scales, as well as transformation to white, unpigmented scales (*Figure 4—figure supplement 5*).

As this locus has been associated with differences in white hindwing margin phenotypes (*Jiggins and McMillan, 1997*; *Figure 1b*), we also targeted *cortex* in mimetic morphs that display the same phenotype in the two species, *H. erato cyrbia* and *H. melpomene cythera*. Mutant scales in these colour pattern morphs were also localised across both wing surfaces, with both white and yellow ectopic scales (*Figures 4c* and *5c*). Both the white and blue colouration in these co-mimics is

structurally derived, indicating that *cortex* loss-of-function phenotype also affects the scale ultra-structure. Furthermore, we observed a positional effect, where ectopic scales in the forewing and anterior compartment of the hindwing shifted to yellow, and posterior hindwing scales became white (*Figure 4c*, *Figure 4—figure supplement 5d*). This positional effect likely reflects differential uptake of the yellow pigment 3-OHK across the wing surface, which may be related to cryptic differential expression of the yellow-white switch *aristaless-1* (*Reed et al., 2008*; *Westerman et al., 2018*).

To further test the conservation of *cortex* function across the *Heliconius* radiation, as well as nymphalids as a whole, we knocked out *cortex* in *H. charithonia* and *H. hecale melicerta*, outgroups to *H. erato* and *H. melpomene,* respectively, and *Danaus plexippus* as an outgroup to all Heliconiini. Again, ectopic yellow and white scales appeared throughout the wing surface in all species, suggesting a conserved function with respect to scale development among butterflies (*Figure 5d–f*, *Figure 5—figure supplements 1–5*).

In summary, *cortex* mKOs appear to not be restricted to any specific wing pattern elements and instead affect regions across the surface of both forewings and hindwings. Mutant scales are always Type I scales, with differing pigmentation (3-OHK, yellow) or structural colouration (white) depending on morph and wing position. The high rate of mosaicism combined with high mortality rates suggests *cortex* is likely developmentally lethal. Mutant clones also appear aggregated, suggesting the *cortex* mKO may affect early phases of cell division or communication during development and produce a growth disadvantage or differential adhesion relative to WT cells that result in grouping effects.

## Nuclear localisation of Cortex protein extends across the wing surface in pupal wings

The *cortex* mRNA expression patterns in larval imaginal disks suggest a dynamic progression in the distal regions, and in a few cases (*Figure 3*; *Nadeau et al., 2016*; *Saenko et al., 2019*), a correlation with melanic patterns whose polymorphisms associate with genetic variation at the *cortex* locus itself. We thus initially hypothesised that like for the *WntA* mimicry gene (*Martin et al., 2012*; *Mazo-Vargas et al., 2017*, *Concha et al., 2019*), the larval expression domains of *cortex* would delimit the wing territories where it is playing an active role in colour patterning. However, our CRISPR based loss-of-function experiments challenge that hypothesis because in all the morphs that we assayed, we found mutant scales across the wing surface.

This led us to re-examine our model and consider that post-larval stages of *cortex* expression could reconcile the observation of scale phenotypes across the entire wing, rather than in limited areas of the wing patterns. To test this hypothesis, we developed a Cortex polyclonal antibody and found nuclear expression across the epithelium of *H. erato demophoon* pupal hindwings without restriction to specific pattern elements (*Figure 6*). In fifth-instar larvae, Cortex protein was detected in a similar pattern to mRNA, with expression visible at the intervein midline of developing wings (*Figure 6a*). Cortex was then detected across the entire wing surface from 24 hr after pupal formation (a.p.f), until 80 hr a.p.f in our time series (*Figure 6b–d*, *Figure 6—figure supplement 1*). Localisation remained nuclear throughout development and appears equal in intensity across hindwing colour pattern elements (*Figure 6e*).

## Modular *cis*-regulatory elements drive the evolution of the mimetic yellow bar

Given the broad effect observed for *cortex* across both wing surfaces, we next tested whether specific expression might be under the control of pattern-specific *cis*-regulatory elements (CREs). In order to look for potential CREs, we performed an Assay for Transposase-Accessible Chromatin using sequencing (*ATAC-seq*) in fifth-instar hindwings of both co-mimetic morphs differing in the presence of the yellow bar in *H. erato* and *H. melpomene* (*Figure 7—source data 1*). We observed many accessible chromatin 'peaks' surrounding *cortex* (*Figure 7—figure supplement 1*), each of which could represent a potentially active CRE. To narrow down candidate peaks that could be regulating *cortex* in a pattern-specific manner, we overlayed association intervals with the *ATAC-seq* signals, which indicate evolved regions between populations of *H. melpomene* differing in the hindwing yellow bar phenotype. Specifically, we applied the phylogenetic weighting strategy Twisst

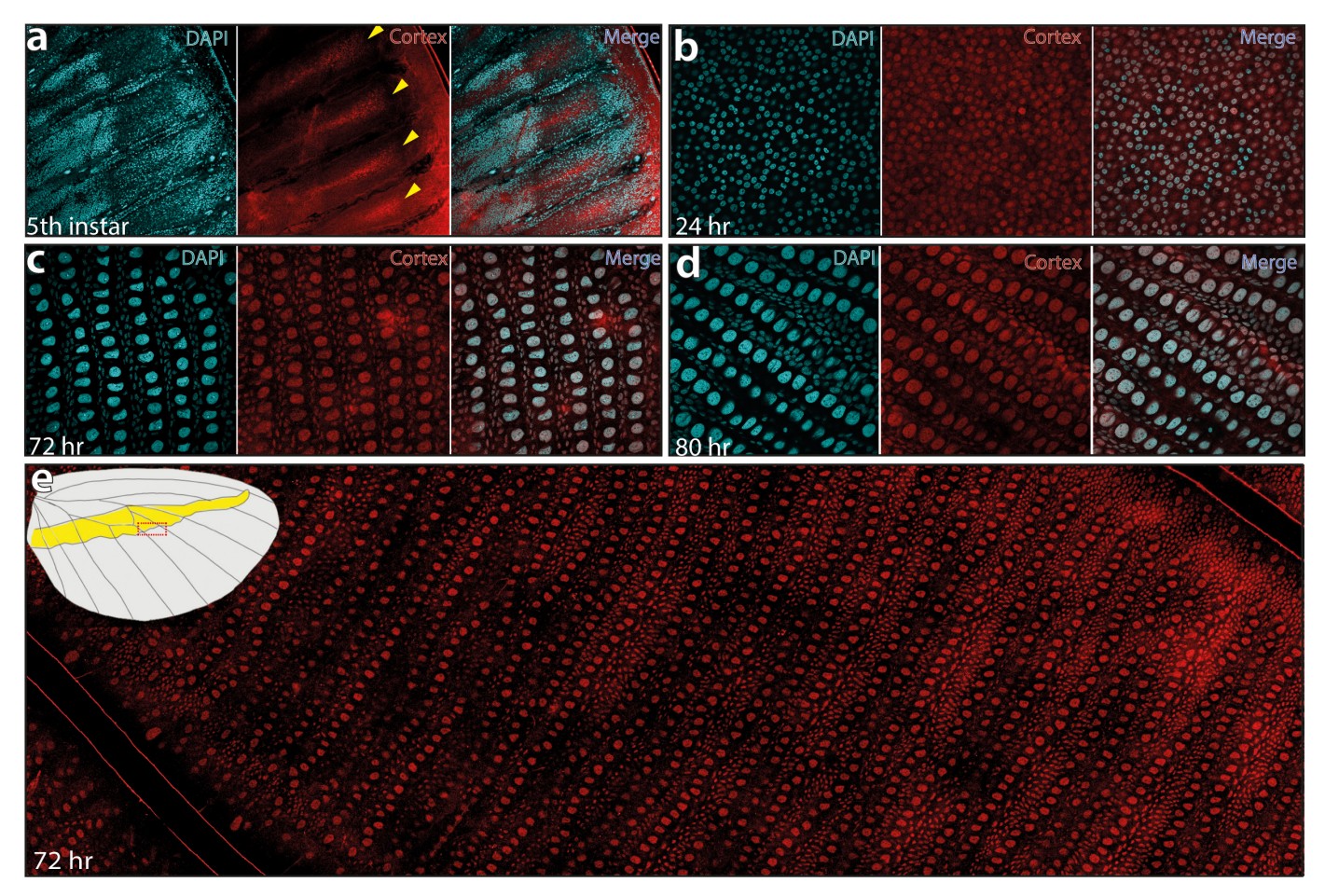

**Figure 6.** Cortex protein localises across the wings in *H. erato demophoon*. Antibody stainings reveal Cortex protein is localised at the distal intervein midline in fifth-instar wing discs (yellow arrowheads) (**a**). At 24 hr APF, the protein is detected across the wing and localised strongly to the cell nuclei (**b**). This localisation continues at 72 hr APF(**c**) and 80 APF (**d**). In each panel, leftmost insert shows nuclei stained with DAPI (magenta), middle insert with Cortex antibody detected with a secondary alexa-fluor 555 conjugated antibody (red), and right insert shows both channels merged into a composite image. No appreciable difference in localisation is detected across presumptive pattern elements at 72 hr APF (**e**). The magnified portion of the imaged wing is indicated in the wing cartoon in the top left corner of ( **e**).

The online version of this article includes the following figure supplement(s) for figure 6:

**Figure supplement 1.** Fifth-instar larval wing disc showing intervein localisation of Cortex protein at 10× (**a**) and 30× magnification of the same area (**a'**).

(topology weighting by iterative sampling of subtrees; *Martin and Van Belleghem, 2017*) to identify shared or conserved genomic intervals between sets of *H. melpomene* populations (obtained from *Moest et al., 2020*) with similar phenotypes around *cortex*. This method identified a strong signal of association ~8 kb downstream of the annotated *cortex* stop codon, that overlapped with a clear *ATAC-seq* peak (*Figure 7a,b*, *Figure 7—figure supplement 1*).

We next sought to knock out this CRE, by designing a pair of sgRNA guides flanking the *ATAC-seq* signal. We reasoned that since *cortex* controlled the switch to melanic scales across the entire wing, knocking-out an enhancer in the melanic morph (*H. melpomene melpomene*), or in F1 hybrids between *H. melpomene rosina* and *H. melpomene melpomene*, should result in the appearance of yellow scales in a yellow bar-like pattern. Indeed, upon KO of this CRE we recovered mKOs consistent with a modular enhancer driving *cortex* expression in a yellow bar-specific pattern, with no clones exhibiting yellow scales extending out of the region that forms the yellow bar (*Figure 7c*, *Figure 7—figure supplement 2*).

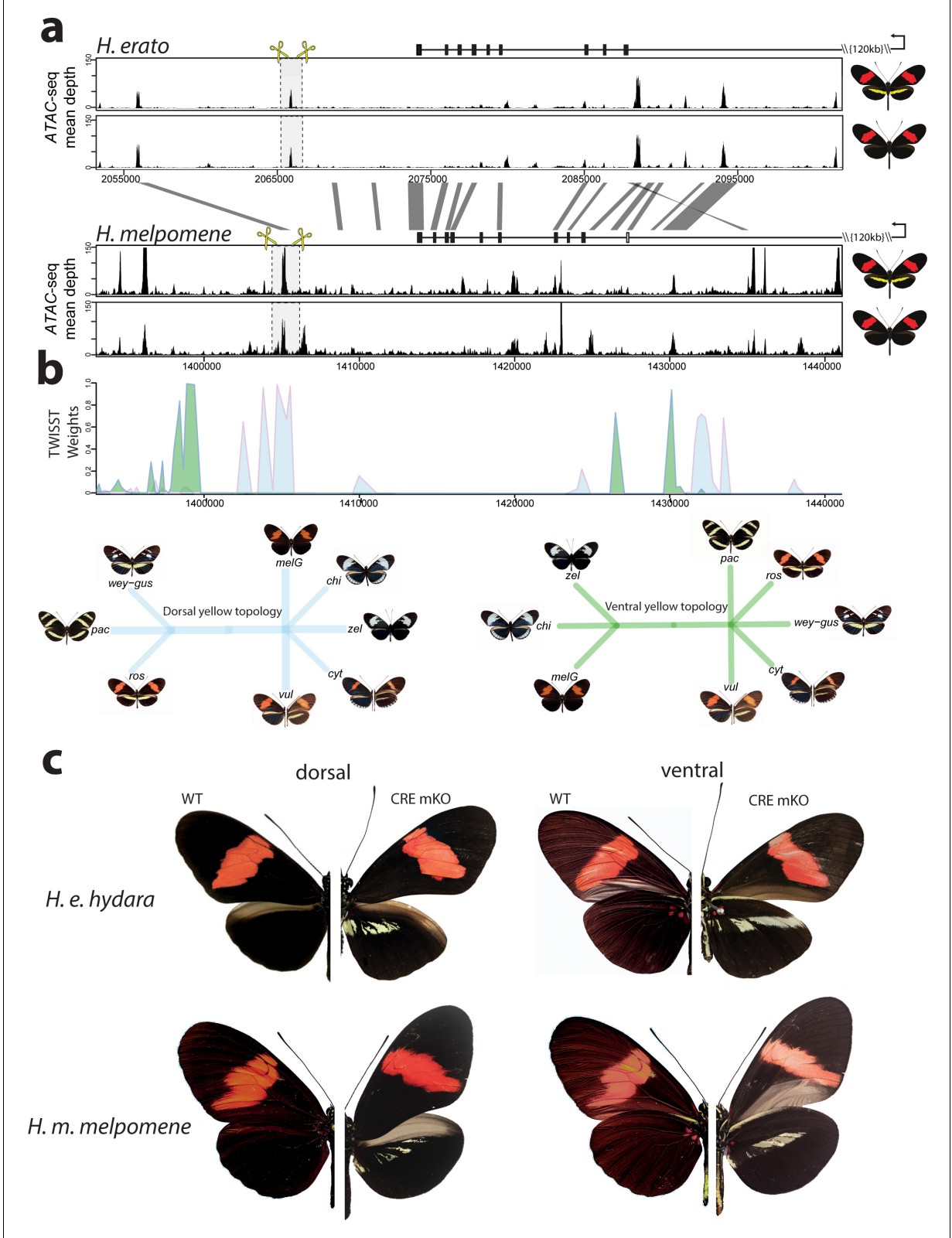

**Figure 7.** Modular CREs control the presence of the yellow band in *Heliconius melpomene* and *Heliconius erato*. (a) Chromatin accessibility as measured by mean sequence depth for *ATAC-seq* traces in *H. erato* (top) and *H. melpomene* (bottom) in fifth-instar caterpillar hindwings in yellow banded and black morphs. The gene model for *cortex* is shown above the traces (black rectangles are coding exons, white rectangle non-coding exon, lines are introns, direction of transcription is indicated by an arrow). The transcription start site is around 120 kb from the first coding exon of *cortex*.
*Figure 7 continued on next page*

*Figure 7 continued*

Positions of sgRNAs used for peak excision are shown (yellow scissors). Regions with >75% sequence identity between *H. melpomene* and *H. erato* are indicated by grey lines. (b) Twisst analysis results showing high genetic association for the presence of a yellow bar in *H. melpomene* populations with a ventral (ventral topology) or dorsal (dorsal topology) yellow bar. Abbreviations for twisst morphs: wey-gus = *H. cydno weymeri-gustavi*, chi = *H. cydno chioneus*, zel = *H. cydno zelinde*, pac = *H. pachinus*, ros = *H. melpomene rosina*, melG = *H. melpomene melpomene* French Guiana, vul = *H. melpomene vulcanus*, cyt = *H. melpomene cythera*. (c) *Cortex* loss-of-function at the yellow bar CREs affect scales only in the presumptive yellow bar region. CRE KO affects both dorsal (left) and ventral (right) hindwings.

The online version of this article includes the following source data and figure supplement(s) for figure 7:

**Source data 1.** List of ATAC-seq samples used in this study, and corresponding accession numbers.
**Figure supplement 1.** The *cortex* locus is characterised by many accessible chromatin peaks as revealed by *ATAC-seq*.
**Figure supplement 2.** *H. melpomene melpomene* wild-type (WT), alongside *cortex* CRE mKO individuals recovered in CRISPR experiments.
**Figure supplement 3.** *H. erato hydara* wild-type (WT), alongside *cortex* CRE mKO individuals recovered in CRISPR experiments.
**Figure supplement 4.** Genotyping at the *H. melpomene* and *H. erato* CRE confirms CRISPR-induced deletions.
**Figure supplement 5.** ATAC-Seq traces at the *cortex* locus for day three old pupal wings in *H. erato lativitta* (a) and *H. erato demophoon* (b).

To test whether the same element was driving the evolution of the yellow bar phenotype in the co-mimetic morph of *H. erato*, we first targeted the homologous peak, which shares both 95% sequence identity with *H. melpomene*, as well as the presence of an accessible chromatin mark (*Figure 7a*). While none of our CRISPR trials resulted in a visible phenotype at this locus (number of injected adults eclosed = 36), we did observe the presence of a further accessible region ~10 kb 3' of the *H. melpomene* conserved CRE. We reasoned that a different but positionally close peak could be driving the yellow bar phenotype in *H. erato*. Remarkably, targeting of this CRE resulted in a yellow bar phenotype in the melanic *H. erato hydara,* with no clones containing yellow scales extending beyond the region that forms the yellow bar (*Figure 7c*, *Figure 7—figure supplement 3*). Deletions at each of the loci were confirmed by PCR amplification, cloning, and Sanger sequencing (*Figure 7—figure supplement 4*).

Finally, to confirm that the CREs were interacting with the *cortex* promoter, we took advantage of a previously published set of Hi-C samples in *H. erato* populations (*Lewis et al., 2019*), to check for enhancer/promoter interactions through the implementation of virtual 4C plots. For both CREs, we found a statistically significant interaction between CRE and promoter, indicating the observed effect is likely due to the CRE interacting with the *cortex* promoter, and not a different gene at the locus (*Figure 7—figure supplement 5*).

## Transposable element insertions are associated with the yellow bar phenotype in geographically distinct *H. melpomene* populations

Given that we were able to induce a yellow bar phenotype by the deletion of a modular CRE, we next asked whether natural populations with this phenotype might also show a similar deletion. To test for the presence of deletions at the candidate CRE, we used extensive publicly available whole-genome re-sequence data for geographically isolated populations differing in the presence of the hindwing yellow bar (*Darragh et al., 2019*; *Enciso-Romero et al., 2017*; *Kozak et al., 2018*; *Martin et al., 2019*; *Van Belleghem et al., 2017*). In total, we assayed 16 geographically isolated subspecies across central and south America and looked for signature coverage drops at the targeted *ATAC-seq* peak, which could be indicative of deletions (*Chan et al., 2010*; *Kemppainen et al., 2021*; *Figure 8—source data 1*). We observed a characteristic drop in coverage at the targeted CRE in all *H. melpomene* and *H. timareta* morphs exhibiting a yellow bar phenotype, while no drop was detected in morphs with a melanic hindwing (*Figure 8a*). Given this characteristic signature associated with the presence of a yellow bar in the sequencing data, we next genotyped across the putative deletion using Sanger sequencing in *H. melpomene rosina* and *H. melpomene melpomene* individuals. Surprisingly, we found two transposable element (TE) insertions in *H. melpomene rosina* with a Helitron-like TE found spanning the CRE peak, suggesting that the coverage drop is instead due to an insertion of repetitive sequence, rather than a deletion. Enhancer disruption is therefore likely caused by TE sequence in yellow bar morphs (*Liu et al., 2019*). We next assayed three other yellow-barred morphs (*H. melpomene bellula*, *H. melpomene amaryllis*, and *H. timareta tristero*) and found the same TE signatures in all three populations (*Figure 8—figure supplement 2*), suggesting the TE insertions are likely shared through introgression. No similar

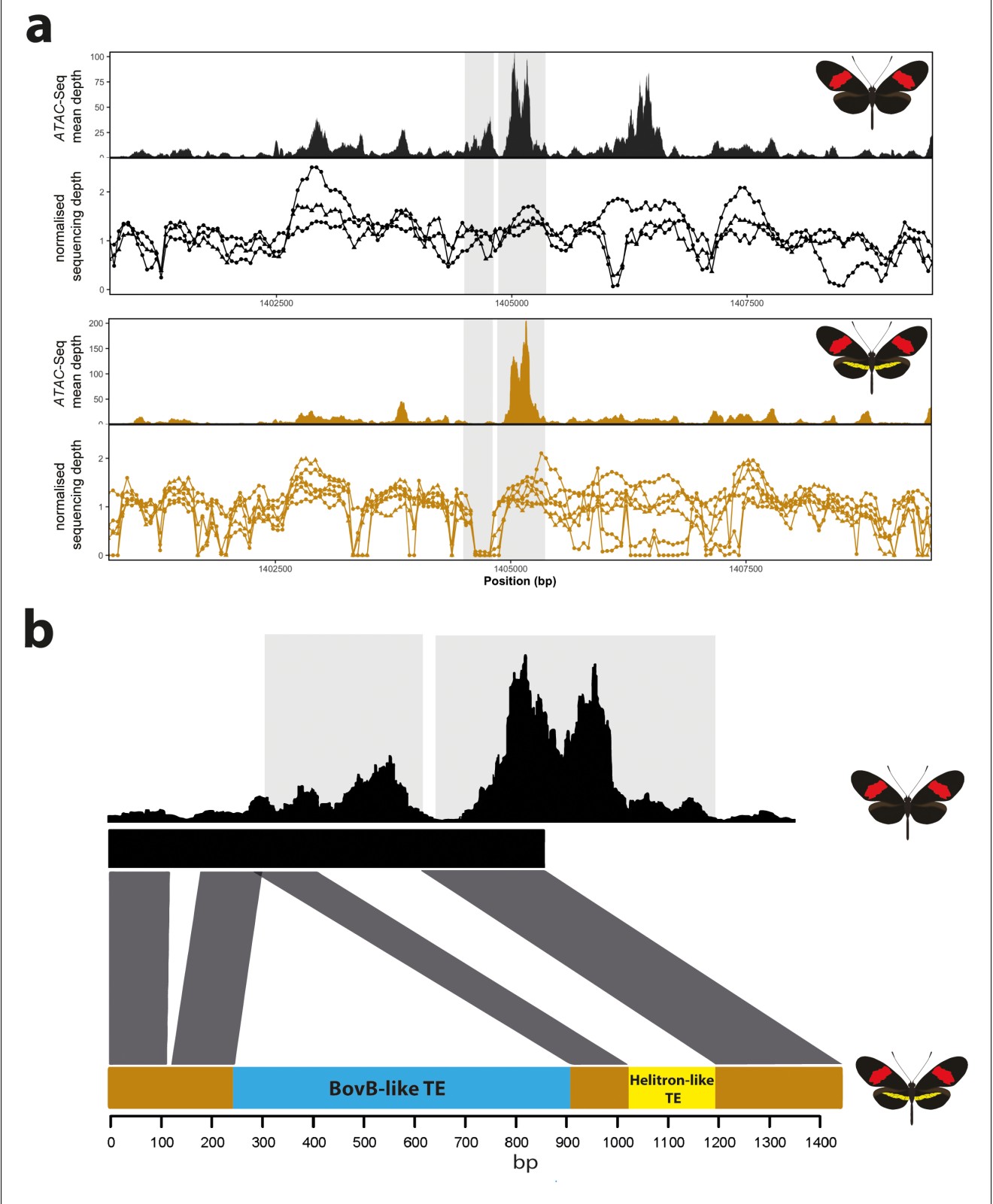

**Figure 8.** Coverage drop indicative of deletions in yellow-barred populations of *H. melpomene*. (a) Mean sequence depth for *ATAC-seq* traces for the excised CRE are shown above normalised depth in sequencing coverage for populations of *H.melpomene* (circles) and H. *timareta* (triangles) differing for the presence of a yellow bar. Yellow-barred populations display a drop in coverage for both ATAC signal and sequencing depth at a position corresponding to a portion of the targeted CRE (highlighted by the grey lines). Morphs analysed for melanic hindwing *H. melpomene* and *H. timareta*:
*Figure 8 continued on next page*

*Figure 8 continued*

*H. m. maletti, H. m. melpomene, H. t. florencia*. Morphs analysed for yellow-barred *H. melpomene* and *H. timareta*: *H. m. bellula, H. m. amaryllis, H. m. rosina, H. m. burchelli, H t. thelxione, H. t. tristero*. (b) Sanger sequencing of target regions in *H. m. melpomene* and *H. m. rosina* reveals an insertion of two TE elements surrounding the yellow bar CRE. A larger BovB-like element of 690 bp (blue) and a smaller 163 bp Helitron-like element (yellow) are present in the *H. m. rosina* sequences, but not the *H. m. melpomene* sequences. Base pair positions of the consensus Sanger sequencing traces are shown below.

The online version of this article includes the following source data and figure supplement(s) for figure 8:

**Source data 1.** List of individuals used in coverage depth analysis, and corresponding accession numbers.
**Source data 2.** Consensus sequences recovered from Sanger sequencing across the *H. melpomene/timareta* CRE.
**Figure supplement 1.** Alignment visualisation of the sequences above.
**Figure supplement 2.** No evidence of deletion at the yellow bar CRE in *H. erato* populations.

signature of reduced coverage was observed in co-mimetic morphs of *H. erato*, suggesting that sequence divergence is responsible for the evolution of the yellow bar CRE in this species (*Figure 8—figure supplement 2*).

## *Cortex* coding KO causes partial homeotic shifts in scale structure

Previous studies have shown an association between scale ultrastructure and pigmentation in *Heliconius* butterflies (*Concha et al., 2019*; *Gilbert et al., 1987*; *Janssen et al., 2001*; *Zhang et al., 2017*). In particular, it has been reported that perturbation by wounding transforms both the pigment content and structure of scales in a tightly coupled way (*Janssen et al., 2001*). We thus asked whether ectopic yellow/white scales generated through *cortex* knockout were accompanied by structural transformations using scanning electron microscopy (SEM) in the same way as ectopic colour scales generated through wounding or *WntA* knockouts (*Janssen et al., 2001*; *Concha et al., 2019*). To account for known positional effects on scale structure, we compared wild-type and mutant scales from homologous locations across the wing surface.

We observed ultrastructural shifts that are consistent with partial homeosis in *cortex* mutant scales in both *H. melpomene* and *H. erato* (*Figure 9*, *Figure 9—figure supplement 1*). In all cases where a yellow or white (Type I) clone was present in a region that would otherwise be black or blue (Type II) in the wild type, the ultrastructure of the scale was notably different. Wild-type blue and black scales have crossribs at a spacing of ~0.6 µm, lack lamina between ridges and crossribs, and have no prominent microribs, while both wild-type and mutant Type I scales have no prominent crossribs, lamina that fills the spaces between the microribs and ridges, and prominent microribs at a spacing of ~0.2 µm (*Figure 9a–d*, *Figure 9—source data 1–2*). A consistent difference between all Type I scales (mutant and wild type) is the presence of a lamina covering the inter-ridge space (*Figure 9f*). These ultrastructural shifts suggest that the perturbation of *cortex* affects scale fate decision, not only shifting pigmentation type, but also scale morphology.

Red scales (Type III) that are within a coding KO clone take on an aberrant structure and pigmentation. Scales were frequently found to be curled up laterally, and while ommochrome pigment is sometimes visibly deposited in the scale, it is granular in appearance rather than diffuse throughout the scale (*Figure 9e*, *Figure 9—figure supplement 2*). These 'granular' pigment accumulations could not be observed as a distinct structure by SEM, suggesting that they are under the surface of the scale. As with wild-type and mutant Type I scales, prominent microribs can also be observed on these rolled scales, but due to the topological deformity of these scales, it was not possible to take accurate measurements.

## Discussion

### *Cortex* is a key scale cell specification gene

The genetic locus containing the gene *cortex* represents a remarkable case of convergent evolution, where repeated and independent mutations surrounding the gene are associated with shifts in scale pigmentation state in at least nine divergent species of Lepidoptera (*Beldade et al., 2009*; *van der Burg et al., 2020*; *Nadeau et al., 2016*; *Van Belleghem et al., 2017*; *VanKuren et al., 2019*; *Van't Hof et al., 2019*; *Van't Hof et al., 2016*). While these studies have linked putative regulatory

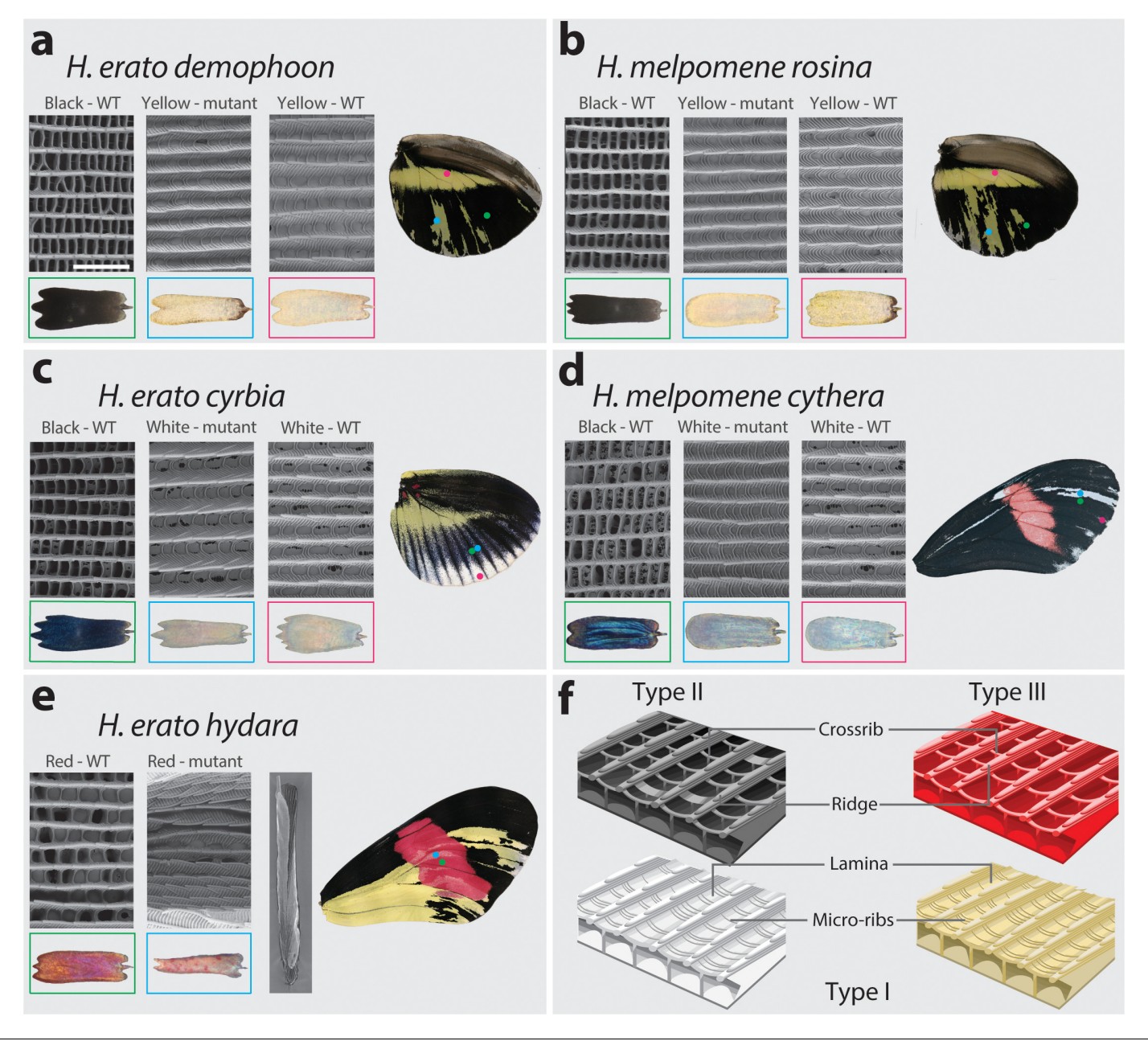

**Figure 9.** SEM reveals structural changes induced by *cortex* KO. Ultrastructural morphology of *H. erato demophoon* (**a**) and *H. melpomene rosina* (**b**) hindwing scales for wild-type (WT) black, mutant yellow and wild-type yellow scales. Light micrographs of each scale are shown below the representative SEM images. Scale morphologies are also presented for morphs displaying shifts to white scales in *H. erato cyrbia* (**c**) and *H. melpomene cythera* (**d**). Structural changes in the red scales of *H. erato hydara* (**e**) are accompanied by scale deformations, resulting in a curled appearance. Cartoon depiction of scale ultrastructure illustrating differences between scale types (**f**). Type I scales are characterised by the presence of a lamina covering the scale windows and by microribs joining the longitudinal ridges. Type II scale cells display larger crossribs and lack a lamina covering the scale windows. Type III scale cells and are characterised by larger spacing between crossribs and ridges. Scale bar in (**a**) = 3 µm.

The online version of this article includes the following source data and figure supplement(s) for figure 9:

**Source data 1.** Pairwise Wilcox test adjusted p-values for quantitative measures of scale structures and features in *H.melpomene*.

**Source data 2.** Pairwise Wilcox test adjusted p-values for quantitative measures of scale structures and features in *H.erato*.

**Figure supplement 1.** Quantitative measures of scale structures and features.

**Figure supplement 2.** *Cortex* KOs in red scales produce aberrant morphologies with features consistent with Type I scale transformations.

variation around *cortex* to the evolution of wing patterns, its precise effect on scale cell identity and pigmentation has remained speculative until now. Here, we demonstrate that *cortex* is a causative gene that specifies melanic and red (Type II and Type III) scale cell identity in *Heliconius* and acts by influencing both downstream pigmentation pathways and scale cell ultrastructure. We also show that *cortex* is under the control of modular enhancers that appear to control the switch between mimetic yellow bar phenotypes in both *H. melpomene* and *H. erato*. Our combination of expression studies and functional knockouts demonstrate that this gene acts as a key scale cell specification switch across the wing surface of *Heliconius* butterflies, and thus has the potential to generate much broader pattern variation than previously described patterning genes.

While we have shown that *cortex* is a key scale cell specification gene, it remains unclear how a gene with homology to the *fizzy/cdc20* family of cell cycle regulators acts to modulate scale identity. In *Drosophila*, Fizzy proteins are known to regulate APC/C activity through the degradation of cyclins, leading to the arrest of mitosis (*Raff et al., 2002*). In particular, *fizzy-related* (*fzr*) induces a switch from the mitotic cycle to the endocycle, allowing the development of polyploid follicle cells in *Drosophila* ovaries (*Schaeffer et al., 2004*; *Shcherbata et al., 2004*). Similarly, *cortex* has been shown to downregulate cyclins during *Drosophila* female meiosis, through its interaction with the APC/C (*Pesin and Orr-Weaver, 2007*; *Swan and Schüpbach, 2007*). Immunostainings show that Cortex protein localises to the nucleus in *Heliconius* pupal wings, suggesting a possible interaction with the APC/C in butterfly scale building cells. Ploidy levels in Lepidoptera scale cells have been shown to correlate with pigmentation state, where increased ploidy and scale size lead to darker scales (*Cho and Nijhout, 2013*; *Iwata and Otaki, 2016*). *cortex* may thus be modulating ploidy levels by inducing endoreplication cycles in developing scale cells. However, we currently have no direct evidence for a causal relationship between ploidy state and pigmentation output, and a mechanistic understanding of this relationship and any role *cortex* may be playing in modulating ploidy levels will require future investigation.

A curious result reported from our *RNA-seq* dataset is that differential expression appears to occur in opposite directions between the two co-mimetic morphs. While this could represent some difference in the precise role of *cortex* between *H. melpomene* and *H. erato*, the pattern may also be explained by the limited sampling during fifth-instar development in both species. The dynamic expression observed during fifth-instar wing development suggests that levels of cortex expression may be changing in a precise and variable manner of short periods of development. A more precise time series across fifth-instar and early pupal development may thus be needed to reveal the precise difference in cellular function of *cortex* between these species.

## The mimetic yellow bar phenotype switch is controlled by the evolution of modular enhancers

In *H.melpomene*, we were able to narrow down a clear peak of association with the presence of accessible chromatin marks and showed that KO of this region results in the appearance of a yellow bar phenotype in black hindwing morphs (*Figure 7*). Interestingly, when targeting the homologous peak in *H. erato*, we failed to recover any type of phenotype, but were able to induce the appearance of a yellow bar through the targeting of an adjacent peak, not present in the *H. melpomene* datasets, indicating that an independently evolved CRE is driving this phenotype in *H. erato*.

These results, coupled with the coding KOs, suggests that the CREs are enhancers that are able to drive *cortex* expression in a yellow bar specific manner. It is therefore puzzling that both the in situ hybridisation and antibody experiments failed to recover an association between Cortex localisation and the yellow bar phenotype. One possibility is that, because *cortex* is expressed throughout the wing, the differences in *cortex* expression that drive the pattern difference are either highly discrete in time and therefore hard to observe, or are the consequence of subtle changes in concentration that we could not detect with immunofluorescence. Moreover, *cortex* is known to have complex patterns of alternative splicing (*Nadeau et al., 2016*), suggesting that perhaps both our polyclonal antibody and in situ probes lack the specificity to detect localisation of specific alternatively spliced variants. This lack of a conspicuous link between expression and function is a puzzling result that will require further investigation in future. The ideal experiments would utilise the identified enhancers as enhancer traps, to show they are able to drive expression in a pattern-specific manner, as well as perform knock-in experiments in the reciprocal co-mimetic morph, to show that these regions are sufficient to drive the phenotypic switches.

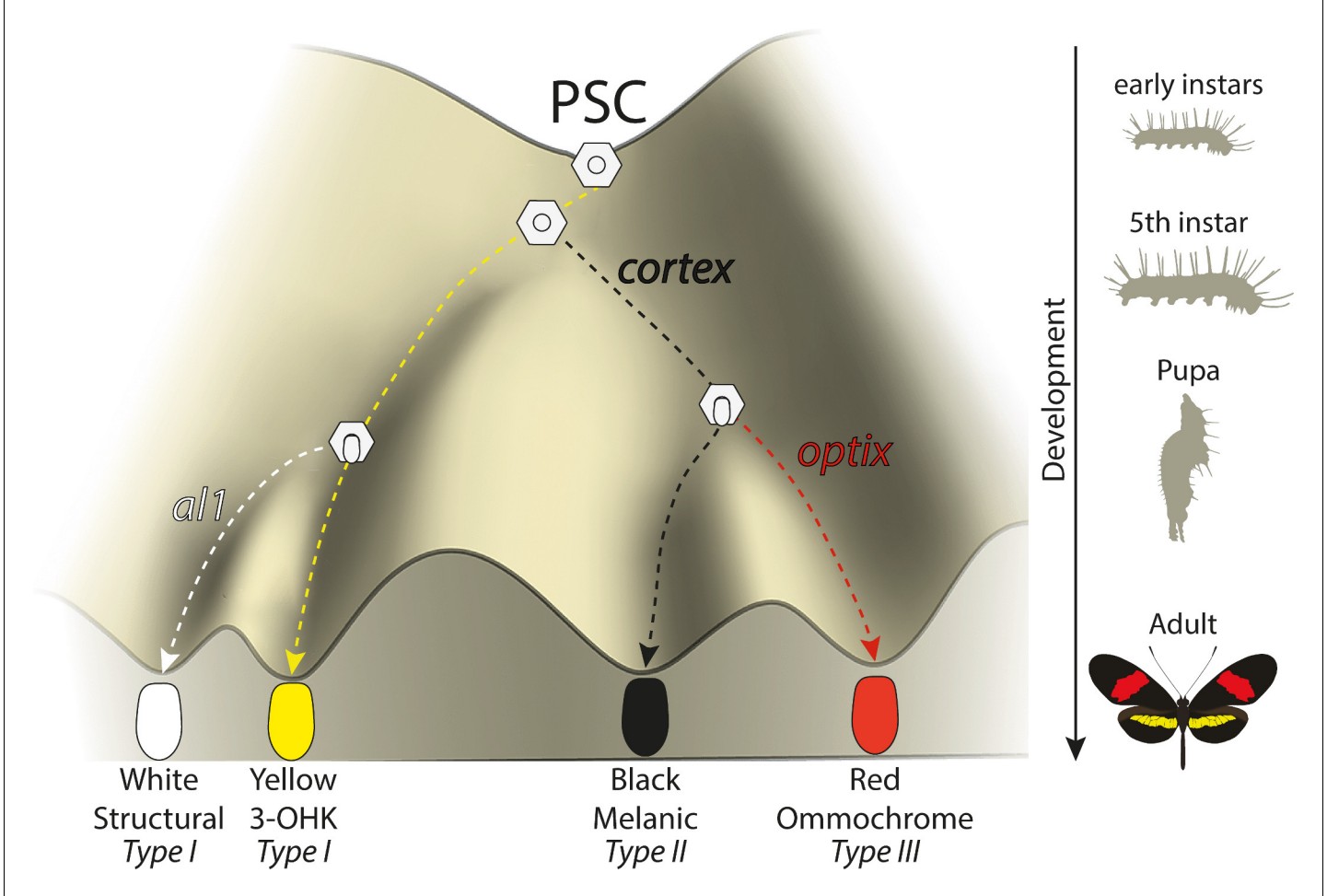

**Figure 10.** Expression of key genes affect scale fate decisions and influence downstream pigmentation state. During early-instar development, wing disc cells differentiate into presumptive scale cells (PSCs). Throughout fifth-instar growth, PSCs express key scale cell specification genes such as *cortex*, which induce differentiation into Type II (*optix−*) scales or Type III (*optix+*) scales. In the absence of *cortex*, scale cells differentiate into Type I scales, which differ in pigmentation state based on 3-OHK synthesis controlled by *aristaless1* expression. Model based on the epigenetic landscape (Waddington) and by observations made by *Gilbert et al., 1987*.

In *H. melpomene*, we found a clear association between the absence of an accessible chromatin peak in yellow-barred populations with a characteristic drop in coverage over the same region, that overlaps with both the targeted CRISPR and association intervals. The mapped profiles show that this drop in coverage is explained by phenotype, rather than geography, in contrast to other adjacent regions. Upon further investigation, we found a large 690 bp TE insertion 5' of the peak of interest as well as a Helitron-like sequence overlapping the peak in *H. melpomene rosina*. This raises the interesting possibility that this portion of the enhancer might contain the binding sites necessary to drive *cortex* in a yellow bar specific manner, and that recurrent TE insertions across this region are driving the evolution of this phenotype in *H. melpomene* populations. We also note that this insertion is observed in mimetic morphs of a different species, *H. timareta*, with which *H. melpomene* has previously been described to share regulatory regions at other patterning loci via adaptive introgression (*Morris et al., 2019*; *Wallbank et al., 2016*). Thus, adaptive introgression of this region and its structural variants is likely facilitating mimicry in this system (*Heliconius Genome Consortium et al., 2012*).

## *Heliconius* wing patterning is controlled by interactions among major patterning genes

Functional knockouts now exist for all the four major loci known to drive pigmentation differences in *Heliconius* (*Mazo-Vargas et al., 2017*; *Westerman et al., 2018*; *Zhang et al., 2017*). These loci represent the major switching points in the GRNs that are ultimately responsible for determining scale cell identity. This work underscores the importance of two patterning loci, *cortex* and *WntA*, as master regulators of scale cell identity. Both are upregulated early in wing development and have broad effects on pattern variation (*Concha et al., 2019*; *Nadeau et al., 2016*). The signalling molecule *WntA* modulates forewing band shape in *Heliconius* by delineating boundaries around patterns elements, and is expressed in close association with future pattern elements (*Concha et al., 2019*; *Martin et al., 2012*). Unlike *cortex* mutants, *WntA* KOs shift scale cell identity to all three cell types (I, II, and III), depending on genetic background. Thus, *WntA* acts as a spatial patterning signal inducing or inhibiting colour in specific wing elements, in contrast to *cortex,* which acts as an 'on-off' switch across all scales on the butterfly wing.

Interestingly, *cortex* knockouts lead to shifts in scale fate irrespective of *WntA* expression. This suggests either that *cortex* is required as an inductive signal to allow *WntA* to signal further melanisation, or that two, independent ways to melanise a scale are available to the developing wing. The latter hypothesis is supported by certain *H. erato* colour pattern *WntA* mutants, where even in putatively *cortex*-positive regions, scales are able to shift to Type I in the absence of *WntA* alone (*Concha et al., 2019*). This indicates that while under certain conditions *cortex* is sufficient to induce the development of black scales, *WntA* is also required as a further signal for melanisation in some genetic backgrounds. Under this scenario, colour pattern morphs may be responding epistatically to different *WntA*/*cortex* alleles present in their respective genetic backgrounds. This is also consistent with genetic evidence for epistasis between these two loci seen in crossing experiments, whereby the yellow bar in *H. erato favorinus* results from an interaction between the *Cortex* and *WntA* loci (Mallet, 1989).

Under a simple model (*Figure 10*), *cortex* is one of the earliest regulators and sets scale differentiation to a specific pathway switches between Type I (yellow/white) and Type II/III (black/red) scales. Thus, we can envision a differentiating presumptive scale cell (PSC) receiving a Cortex input as becoming Type II/III competent, with complete Type III differentiation occurring in the presence of *optix* expression (*Zhang et al., 2017*). This is consistent with our data, which shows *cortex* is also required as a signal for Type III (red) scales to properly develop. Several *cortex* mutant individuals had clones across red pattern elements and failed to properly develop red pigment. The development of red scales in *Heliconius* butterflies is also dependent on expression of the transcription factor *optix* during mid-pupal development (*Lewis et al., 2019*; *Reed et al., 2011*; *Zhang et al., 2017*). Therefore, *cortex* expression is required for either downstream signalling to *optix* or to induce a permissive scale morphology for the synthesis and deposition of red pigment in future scales. *Cortex* is thus necessary for the induction of Type III scale cells but insufficient for their proper development.

Conversely, a PSC lacking a Cortex input differentiates into a Type I scale, whose pigmentation state depends on the presence of the transcription factor *aristaless1* (*al1*), where *al1* is responsible for inducing a switch from yellow to white scales in *Heliconius* by affecting the deposition of the yellow pigment 3-OHK (*Westerman et al., 2018*). The uptake of 3-OHK from the haemolymph occurs very late in wing development, right before the adult ecloses (*Reed et al., 2008*). Our *cortex* mKOs revealed a shift to both yellow and white scales, with their appearance being positionally dependent; more distally located scales generally switch to white, while more proximal scales become yellow. This pigmentation state is likely controlled by differences in *al1* expression varying between wing sections in different ways across morphs.

However, the switch induced by Cortex under this model is likely not a simple binary toggle, and is perhaps dependent on a given protein threshold or heterochrony in expression rather than presence/absence, as we find that Cortex protein also localises to the presumptive yellow bar in developing pupal wings. Moreover, the *RNA-seq* data presented suggests other linked genes may also be playing a role in controlling pattern switches between *Heliconius* morphs. In particular, we report the presence of a bi-cistronic transcript containing the ORFs of the genes *dome/wash*, which are differentially expressed during early pupal wing development. While a precise role for *dome/wash* in

wing patterning remains to be demonstrated, it raises the possibility that multiple linked genes co-operate during *Heliconius* wing development to drive pattern diversity. It is noteworthy that in the locally polymorphic *H. numata*, all wing pattern variation is controlled by inversions surrounding *cortex* and *dome/wash*, both of which are also differentially expressed in *H. numata* (*Saenko et al., 2019*). This raises the interesting possibility that evolution has favoured the interaction of multiple genes at the locus that have since become locked into a supergene in *H. numata*.

## Conclusions

The utilisation of 'hotspots' in evolution has become a recurring theme of evolutionary biology, with several examples in which independent mutations surrounding the same gene have driven adaptive evolution (e.g *Pitx1*, *Scute*) (*Stern and Orgogozo, 2009*). One proposed facilitator of such hotspots is through the action of genes acting as 'input-output' modules, whereby complex spatio-temporal information is translated into a co-ordinated cell differentiation program, in a simple switch-like manner. One prediction of the nature of such genes would be a switch-like behaviour such as that observed for *cortex* in this study, as well as the presence of a complex modular *cis*-regulatory architecture surrounding the gene that is able to integrate the complex upstream positional information into the switch-like output. A conserved feature of the *cortex* locus in Lepidoptera is the presence of large intergenic regions surrounding the gene, with evidence these may be acting as modular *cis*-regulatory switches in *Heliconius* (*Enciso-Romero et al., 2017*; *Van Belleghem et al., 2017*), fitting the predicted structure of input-output genes. Unlike canonical input-output loci however, *cortex* expression appears not to be restricted to any particular colour pattern element in any given species/morph, and yet is capable of producing a switch-like output (Type I vs Type II/III scales). Furthermore, our work shows that two independent CREs in *H. melpomene* and *H. erato* evolved to control the presence/absence of a yellow hindwing bar. However, it is still unclear how *cortex* mechanistically affects pigmentation differences, and given its widespread usage throughout Lepidoptera, it is of general interest to understand its role in driving scale pigmentation.

# Materials and methods

## Butterfly husbandry

*Heliconius* butterflies were collected in the tropical forests of Panama and Ecuador. Adults were provided with an artificial diet of pollen/glucose solution supplemented with flowers of *Psiguria*, *Lantana*, and/or *Psychotria alata* according to availability. Females were provided with Passiflora plants for egg laying (*P. menispermifolia* for *H. melpomene*, *P. biflora* for *H. erato* and *H. charithonia*, and *P. vitifolia* for *H. hecale*). Eggs were collected daily, and caterpillars reared on fresh shoots of *P. williamsi* (*melpomene*), *P. biflora* (*erato* and *charithonia*), and *P. vitifolia* for *H. hecale*. Late fifth (final) instar caterpillars were separated into individual pots in a temperature-monitored room for *RNA-seq* experiments, where they were closely observed for the purpose of accurate developmental staging.

## Phylogenetic analysis of *domeless* and *cortex*

To identify orthologues of *dome* across the Lepidoptera, we performed tBLASTn searches using the previously annotated *H. melpomene* Hmel2 (Hm) and *H.erato demophoon* V1 (Hed) *dome* sequences against the genomes of *Operophtera brumata* V1 (Ob), *Trichoplusia ni* Hi5.VO2 (Tn), *Bombyx mori* ASM15162v1 (Bm), *Manduca sexta* 1.0 (Ms), *Plodia interpunctella* V1 (Pi), *Amyeolis transitella* V1 (At), *Phoebis sennae* V1.1 (Ps), *Bicyclus anynana* V1.2 (Ba), *Danaus plexippus* V3 (Dp), *Dryas iulia* helico3 (Di), *Agraulis vanillae* helico3 (Av), *Heliconius erato lativitta* V1 (Hel) genomes found on Lepbase (*Challi et al., 2016*). As a trichopteran outgroup, we used a recently published Pacbio assembly of *Stenopsyche tienmushanensis* (St) (*Luo et al., 2018*). Recovered amino acid translations were aligned using clustal omega (*Madeira et al., 2019*). The resulting alignments were used to produce a phylogenetic tree using PhyML (*Guindon et al., 2010*), based on a best fit model using AIC criterion (selected model was JTT + G + I+F). The tree was visualised and re-rooted to the Trichopteran outgroup using FigTree.

To confirm *cortex* as a cdc20 gene, we retrieved full-length protein homologs from TBLASTN searches and used them to generate a curated alignment with MAFFT/Guidance2 with a column threshold of 0.5. Guidance2 is an alignment reliability method that parses aligned residues while

also maximising tree robustness (*Sela et al., 2015*), thus ruling out biases introduced from paralog-specific domains. We then constructed a maximum-likelihood tree using W-IQ-TREE with the 'Auto' function to find a best-fit model of substitution.

## Tissue sampling and *RNA-seq*

*H. melpomene rosina* and *H. erato demophoon* butterflies were collected around Gamboa, Panama; *H. melpomene melpomene* and *H. erato hydara* butterflies were collected around Puerto Lara, Darien, Panama. Methodology for sample preparation and sequencing was performed as previously descri *Hanly et al., 2019*. The datasets generated and/or analysed during the current study are available in the SRA repository (PRJNA552081). Reads from each species were aligned to the respective genome assemblies Hmel2 (*Davey et al., 2016*) and Herato_demophoon_v1 (*Van Belleghem et al., 2017*), using Hisat2 with default parameters (*Kim et al., 2019*). The genomes and annotations used are publicly available at http://www.lepbase.org. Reads were counted with HTSeq-count in union mode (Anders et al., 2015) and statistical analysis performed with the R package DESeq2 (*Love et al., 2014*). Comparisons for larvae were for whole hindwings, grouping samples by pattern form. Samples for pupal stages included wings that were dissected into anterior and posterior compartment as in *Hanly et al., 2019*, and were analysed in DESeq2 using the GLM:

$$\sim individual + compartment^*morph$$

(Compartments: anterior hindwing [HA], posterior hindwing [HPo]). *H. melpomene* and *H. erato* were analysed separately; homology between genes was determined by reciprocal BLAST. The fold-changes and adjusted p-values given in *Figure 2* reflect the primary contrast, showing the effect of pattern form given the effect of compartment. Read counts were determined for whole hindwings at all stages.

## RT-qPCR

The expression level of *cortex* in larval hindwings was further analysed by qPCR in *H. e. demophoon* and *H. e. hydara*. Three individuals were used for each morph. Each individual was an independent replicate (i.e. no pooling of samples). RNA was extracted from the hindwing tissue of larva using Trizol followed by DNase-treatment. An mRNA enrichment was performed using the Dynabeads mRNA purification kit (Thermo Fisher). The mRNA was then converted to cDNA by reverse transcription using the iScript cDNA synthesis kit (Bio-Rad). All reactions had a final cDNA concentration of 2 ng $\mu l^{-1}$ and a primer concentration of 400 nM. The RT-qPCR was carried out using Brilliant III Ultra-fast SYBR green qPCR master mix (Agilent Technologies), on a AriaMx Real-time PCR system (Agilent Technologies) according to manufacturer's instructions. The PCR programme consisted of 95℃ for 2 min followed by 40 cycles of 95℃ for 5 s, 58℃ for 30 s, and 70℃ for 30 s. qPCR experiments were performed using three biological replicates, three technical replicates, and a no template control. Expression levels were normalised using the geometric mean of three housekeeping genes, *eF1α*, *rpL3* and *polyABP* that have previously been validated for *Heliconius numata* (*Piron Prunier et al., 2016*). The relative expression levels were analysed using the R = $2^{-\Delta\Delta Ct}$ method (*Livak and Schmittgen, 2001*). Primer specificity was confirmed using melting curve analysis and the PCR products were checked on a 2% (w/v) agarose gel. The primer efficiency of each gene was calculated using the standard curve given by a 10-fold serial dilution of cDNA (1, $10^{-1}$, $10^{-2}$, $10^{-3}$, $10^{-4}$) and regression coefficient ($R^2$) values.

## In situ hybridisations

Fifth-instar larval wing disks and whole mount in situ hybridisations were performed following a published procedure (*Martin and Reed, 2014*) and imaged using a Leica S4E microscope (Leica Microsystems). Riboprobe synthesis was performed using the following primers from a fifth-instar wing disc cDNA library extracted from *H. melpomene*:

Forward primer 5'-CCCGAGATTCTTTCAGCGAAAC-3' and Reverse primer 5'- ACCGCTCCAA-CACCAAGAAG-3'. Templates for riboprobes were designed by attaching a T7 promoter through PCR and performing a DIG labelled transcription reaction (Roche). The same *H. melpomene* probe was used in all in situ hybridisation experiments. The resulting probe spanned from Exon two to Exon 7 and was 841 bp long.

## Immunohistochemistry and image analysis

Pupal wings were dissected around 60–70 hr post-pupation in PBS and fixed at room temperature with fix buffer (400 µl 4% paraformaldehyde, 600 µl PBS 2 mM EGTA) for 30 min. Subsequent washes were done in wash buffer (0.1% Triton-X 100 in PBS) before blocking the wings at 4℃ in block buffer (0.05 g bovine serum slbumin, 10 ml PBS 0.1% Triton-X 100). Wings were then incubated in primary antibodies against Cortex (1:200, monoclonal rabbit anti-Cortex) at 4℃ overnight, washed, and added in secondary antibody (1:500, donkey anti-rabbit lgG, AlexaFlour 555, Thermo-Fisher Scientific A-31572). Before mounting, wings were incubated in DAPI with 50% glycerol overnight and finally transferred to mounting medium (60% glycerol/40% PBS 2 mM EGTA) for imaging. Z-stacked two-channelled confocal images were acquired using a Zeiss Cell Observer Spinning Disk Confocal microscope.

## CRISPR/Cas9 genome editing

Guide RNAs were designed corresponding to $GGN_{20}NGG$ sites located within the *cortex* coding region and across putative CREs using the program Geneious (*Kearse et al., 2012*). To increase target specificity, guides were checked against an alignment of both *H. melpomene* and *H. erato* resequence data at the scaffolds containing the *cortex* gene (*Moest et al., 2020*; *Van Belleghem et al., 2017*), and selected based on sequence conservation across populations. Based on these criteria, each individual guide was checked against the corresponding genome for off-target effects, using the default Geneious algorithm. Guide RNAs with high conservation and low off-target scores were then synthesised following the protocol by *Bassett and Liu, 2014*. Injections were performed following procedures described in *Mazo-Vargas et al., 2017*, within 1–4 hr of egg laying. Several combinations of guide RNAs for separate exons at different concentrations were used for different injection experiments. For *H. charithonia*, we used the *H. erato*-specific guides, and for *H. hecale*, we used the *H. melpomene* guides.

## Genotyping

DNA was extracted from mutant leg tissue and amplified using oligonucleotides flanking the sgRNAs target region. PCR amplicons were column purified, subcloned into the pGEM-T Easy Vector System (Promega), and sequenced on an ABI 3730 sequencer.

## ATAC-seq

*H. melpomene rosina* and *H. erato demophoon* butterflies were collected around Gamboa, Panama; *H. melpomene melpomene* and *H. erato hydara* butterflies were collected around Puerto Lara, Darien, Panama. Caterpillars of each species were reared on their respective host plants and allowed to grow until the wandering stage at fifth instar. Live larvae were placed on ice for 1–2 min and then pinned and dissected in 1× ice cold PBS. The colour of the imaginal discs, as well as the length of the lacunae, gradually change throughout the larva's final day of development, so that they can be used to confirm the staging inferred from pre-dissection cues (*Reed et al., 2008*). All larvae used for this project were stage 3.5 or older. *ATAC-seq* protocol was based on previously described methodology (*Lewis and Reed, 2019*) and edited as follows. The imaginal discs were removed and suspended whole in 350 µl of sucrose solution (250 mM D-sucrose, 10 mM Tris–HCl, 1 mM. $MgCl_2$, 1× protease inhibitors [Roche]) inside labelled 2 ml dounce homogenisers (Sigma-Aldrich) for nuclear extraction. Imaginal discs corresponding to the left and right hindwings were pooled. After homogenising the tissue on ice, the resulting cloudy solution was centrifuged at 1000 rcf for 7 min at 4 ℃. The pellet was then resuspended in 150 µl of cold lysis buffer (10 mM Tris–HCl, 10 mM NaCl, 3 mM $MgCl_2$, 0.1% IGEPAL CA-630 [Sigma-Aldrich], 1× protease inhibitors) to burst the cell membranes and release nuclei into the solution. Samples were then checked under a microscope with a counting chamber following each nuclear extraction, to confirm nuclei dissociation and state and to assess the concentration of nuclei in the sample. Finally, based on these observations, a calculation to assess number of nuclei, and therefore DNA, to be exposed to the transposase was performed. This number was fixed on 400,000 nuclei, which is the number of nuclei with ~0.4 Gb genomes (*H. erato* genome size) required to obtain the amount of DNA for which *ATAC-seq* is optimised (*Buenrostro et al., 2013*). For *H. melpomene*, this number was 500,333, since the genome size of *H. melpomene* is 0.275 Gb. (*Buenrostro et al., 2013*). For quality control, a 15 µl aliquot of nuclear

suspension was stained with trypan blue, placed on a hemocytometer and imaged at 64×. After confirmation of adequate nuclear quality and assessment of nuclear concentration, a subsample of the volume corresponding to 400,000 nuclei (*H. erato*) and 500,333 (*H. melpomene*) was aliquoted, pelleted 1000 rcf for 7 min at 4°C and immediately resuspended in a transposition mix, containing Tn5 in a transposition buffer. The transposition reaction was incubated at 37°C for exactly 30 min. A PCR Minelute Purification Kit (Qiagen) was used to interrupt the tagmentation and purify the resulting tagged fragments, which were amplified using custom-made Nextera primers and a NEBNext High-fidelity 2× PCR Master Mix (New England Labs). Library amplification was completed between the STRI laboratory facilities in Naos (Panama) and Cambridge (UK). The amplified libraries were sequenced as 37 bp paired-end fragments with NextSeq 500 Illumina technology at the Sequencing and Genomics Facility of the University of Puerto Rico.

## Topology weighting by iterative sampling of subtrees (Twisst) analysis

We applied the phylogenetic weighting strategy Twisst (topology weighting by iterative sampling of subtrees; *Martin and Van Belleghem, 2017*) to identify shared or conserved genomic intervals between sets of *Heliconius melpomene* and *Heliconius cydno* populations with similar phenotypes around the *cortex* gene locus on chromosome 15. Given a tree and a set of pre-defined groups Twisst determines a weighting for each possible topology describing the relationship of groups or phylogenetic hypothesis. Similar to *Enciso-Romero et al., 2017* we evaluated the support for two alternative phylogenetic hypotheses using genomic data obtained from *Moest et al., 2020*. Hypothesis one tested for monophyly of samples that have a dorsal yellow hindwing bar. This comparison included the geographic colour patterns morphs with a dorsal hindwing bar *H. m. rosina*, *H. c. weymeri weymeri*, *H. c. weymeri gustavi* and *H. pachinus* versus the all-black dorsal hindwing morphs *H. m. vulcanus*, *H. m. melpomene* (French Guiana), *H. m. cythera*, *H. c. chioneus* and *H. c. zelinde*. Hypothesis two tested for monophyly of samples that have a ventral yellow hindwing bar versus an all-black ventral hindwing. This comparison included the geographic colour patterns morphs with a ventral hindwing bar *H. m. rosina*, *H. m. vulcanus*, *H. m. cythera*, *H. c. weymeri weymeri*, *H. c. weymeri gustavi* and H. *pachinus* versus the all-black ventral hindwing morphs *H. m. melpomene* (French Guiana), *H. c. chioneus* and *H. c. zelinde*. Maximum-likelihood trees were built from sliding windows of 50 SNPs with a step size of 20 SNPs using PhyML v3.0 (*Guindon et al., 2010*) and tools available at https://github.com/simonhmartin/twisst (*Martin, 2020*).Only windows were considered that had at least 10 sites for which each population had at least 50% of its samples genotyped. Twisst was run with a fixed number of 1000 subsampling iterations.

## Hi-C and virtual 4C plots

Analysis of chromatin contacts between distal cis-regulatory loci and the *cortex* promoter region was performed as previously described (*Lewis et al., 2019* PNAS, *Lewis and Van Belleghem, 2020*). In brief, Hi-C data produced from day three pupal *H. e. lativitta* wings was used to generate an empirical expected distribution and read counts for Hi-C contacts between 5 kb windows centred on the distal CRE and *cortex* promoter were used to determine the observed contacts between loci. Fisher's exact test was then performed to determine significance of the observed contacts relative to those expected from the background model. Virtual Hi-C signal plots were generated using a custom python script (*Ray et al., 2019*).

## Coverage depth analysis and TE genotyping

High-depth whole-genome sequences of 16 *H. m*elpomene, *H. timareta*, and *H. erato* subspecies were obtained from the European Nucleotide Archive, accession numbers can be found in *Figure 8—source data 1*. To assess structural variation putatively affecting the yellow phenotype, reads were mapped to the reference genomes of subspecies that lacked the yellow bar stored in the genome browser Lepbase, 'hmel2.5' for *H. melpomene* and *H. timareta*, and 'Heliconius_erato_lativitta_v1' for *H. erato* (*Challi et al., 2016*) with BWA mem (*Li and Durbin, 2009*). Median sequencing depths across the scaffold containing cortex were computed for all individuals (n = 79) in 50 bp sliding windows and a mapping quality threshold of 30 with the package Mosdepth (*Pedersen and Quinlan, 2018*). Window median depths were normalised by dividing them by the mean depth for the full scaffold per individual. We then averaged the normalised median depths of all individuals per

subspecies, to visualise deviations from the mean sequencing depth across the region in geographically widespread subspecies with and without the yellow bar. We then genotyped across the putative *H. melpomene* deletion using the primers employed for CRISPR genotyping (See *Figure 4— source data 2*). The products were then cloned into the pGEM-T Easy Vector System (Promega) and sequenced them on an ABI 3730 sequencer from both directions using T7 forward and M13 reverse primers. Sequencing was performed from three separate colonies, and a consensus sequence was created based on an alignment of the three replicates from populations of *H. m. melpomene* and *H. m. rosina*.

## SEM imaging

Individual scales from wild-type and mutant regions of interest were collected by brushing the surface of the wing with an eyelash tool, then dusted onto an SEM stub with double-sided carbon tape. Stubs were then colour imaged under the Keyence VHX-5000 microscope for registration of scale type. Samples were sputter-coated with one 12.5 nm layer of gold for improving sample conductivity. SEM images were acquired on a FEI Teneo LV SEM, using secondary electrons and an Everhart-Thornley detector using a beam energy of 2.00 kV, beam current of 25 pA, and a 10 μs dwell time. Individual images were stitched using the Maps 3.10 software (ThermoFisher Scientific).

## Morphometrics analysis

Morphometric measurements of scale widths and ridge distances were carried out on between 10 and 20 scales of each type, using a custom semi-automated R pipeline that derives ultrastructural parameters from large SEM images (*Day et al., 2019*). Briefly, ridge spacing was assessed by Fourier transforming intensity traces of the ridges acquired from the FIJI software (*Schindelin et al., 2012*). Scale width was directly measured in FIJI by manually tracing a line, orthogonal to the ridges, at the section of maximal width.

## Acknowledgements

We thank Oscar Paneso, Elizabeth Evans, Rachel Crisp, and Cruz Batista, for technical support with rearing of butterflies and CRISPR larvae, and to Markus Möest, and Tim Thurman for assistance with butterfly collection. We are also grateful to Krzysztof 'Chris' Kozak and Chi Yun for thoughtful discussions and feedback on the manuscript. We thank the GW Nanofabrication and Imaging Center for enabling SEM, and in particular Christine Brantner, Chris Day, and Anastas Popratiloff for their technical assistance.

## Additional information

### Funding

| Funder | Grant reference number | Author |
|---|---|---|
| Biotechnology and Biological Sciences Research Council | BB/R007500/1 | Luca Livraghi<br>Eva SM van der Heijden<br>Ian A Warren<br>Charlotte Wright<br>Chris D Jiggins |
| National Science Foundation | IOS-1656553 and IOS-1755329 | Joseph J Hanly<br>Ling Sheng Loh<br>Anna Ren<br>Arnaud Martin |
| Puerto Rico Science, Technology and Research Trust | #2020-00142 | Steven M Van Bellghem<br>Riccardo Papa |
| Smithsonian Institution | NSF IOS-1656389 | Carolina Concha<br>W Owen McMillan |
| National Science Foundation | NSF IOS 1656389 | Riccardo Papa |

The funders had no role in study design, data collection and interpretation, or the decision to submit the work for publication.

## Author contributions
Luca Livraghi, Conceptualization, Resources, Data curation, Formal analysis, Supervision, Validation, Investigation, Visualization, Methodology, Writing - original draft, Project administration, Writing - review and editing; Joseph J Hanly, Resources, Data curation, Software, Formal analysis, Validation, Investigation, Visualization, Methodology, Writing - review and editing; Steven M Van Bellghem, Data curation, Software, Formal analysis, Validation, Investigation, Visualization, Methodology, Writing - review and editing; Gabriela Montejo-Kovacevich, Data curation, Formal analysis, Validation, Visualization, Methodology, Writing - review and editing; Eva SM van der Heijden, Formal analysis, Investigation, Writing - review and editing; Ling Sheng Loh, Zachary H Goldberg, Formal analysis, Investigation, Methodology; Anna Ren, Formal analysis, Investigation, Visualization, Methodology; Ian A Warren, Investigation, Methodology, Project administration; James J Lewis, Formal analysis, Validation, Visualization, Methodology; Carolina Concha, Jonah M Walker, Jessica Foley, Michael W Perry, Formal analysis, Investigation; Laura Hebberecht, Data curation, Formal analysis, Investigation, Methodology; Charlotte J Wright, Data curation, Investigation, Methodology; Henry Arenas-Castro, Investigation; Camilo Salazar, Resources, Data curation; Riccardo Papa, Supervision, Funding acquisition, Writing - review and editing; Arnaud Martin, Conceptualization, Resources, Formal analysis, Funding acquisition, Validation, Investigation, Visualization, Methodology, Writing - review and editing; W Owen McMillan, Chris D Jiggins, Conceptualization, Resources, Supervision, Funding acquisition, Validation, Project administration, Writing - review and editing

## Author ORCIDs
Luca Livraghi (ID) https://orcid.org/0000-0002-2597-7550
Joseph J Hanly (ID) https://orcid.org/0000-0002-9459-9776
Steven M Van Bellghem (ID) https://orcid.org/0000-0001-9399-1007
Gabriela Montejo-Kovacevich (ID) https://orcid.org/0000-0003-3716-9929
Ling Sheng Loh (ID) https://orcid.org/0000-0003-0981-7984
Charlotte J Wright (ID) https://orcid.org/0000-0002-3971-4372
Jonah M Walker (ID) https://orcid.org/0000-0001-7355-3130
Jessica Foley (ID) https://orcid.org/0000-0002-4566-3989
Zachary H Goldberg (ID) https://orcid.org/0000-0003-2972-9682
Henry Arenas-Castro (ID) https://orcid.org/0000-0003-4845-9999
Camilo Salazar (ID) https://orcid.org/0000-0001-9217-6588
Michael W Perry (ID) https://orcid.org/0000-0002-5977-8031
Arnaud Martin (ID) https://orcid.org/0000-0002-5980-2249
Chris D Jiggins (ID) https://orcid.org/0000-0002-7809-062X

## Decision letter and Author response
Decision letter https://doi.org/10.7554/eLife.68549.sa1

# Additional files

## Supplementary files
- Transparent reporting form

## Data availability
ATAC-Seq sequencing data have been deposited under ENA BioProject (accession number PRJEB43672). Raw data on morphometrics and high magnification images of mutants are available on Dryad (doi:https://doi.org/10.5061/dryad.8gtht76m0).

The following datasets were generated:

| Author(s) | Year | Dataset title | Dataset URL | Database and Identifier |
|---|---|---|---|---|
| Livraghi L, Hanly JJ, | 2021 | Cortex cis-regulatory switches | https://doi.org/10.5061/ | Dryad Digital |

| Martin A | | | establish scale colour identity and pattern diversity in Heliconius | dryad.8gtht76m0 | Repository, 10.5061/dryad.8gtht76m0 |
| --- | --- | --- | --- | --- | --- |
| Livraghi L, Hanly JJ, VanBelleghem SM, Montejo-Kovacevich G, Heijden SME, Sheng LL, Ren A, Warren IA, Lewis JJ, Concha C, Lopez LH, Charlotte W, Walker MW, Foley J, Goldberg HZ, Arenas-Castro H, Salazar C, Perry WM, Riccardo P, Arnaud M, McMillan WO, Jiggins CD | | 2021 | Cortex cis-regulatory switches establish scale colour identity and pattern diversity in Heliconius | https://www.ebi.ac.uk/ena/browser/view/PRJEB43672 | ENA, PRJEB43672 |

The following previously published dataset was used:

| Author(s) | Year | Dataset title | Dataset URL | Database and Identifier |
| --- | --- | --- | --- | --- |
| Hanly, JJ, Wallbank RWR, Jiggins CDM, cMillan WO | 2019 | Wing RNAseq from Heliconius melpomene, Heliconius erato, Agraulis vanillae Raw sequence reads | https://www.ncbi.nlm.nih.gov/bioproject/PRJNA552081/ | NCBI BioProject, SRAPRJNA552081 |

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
