## [Decision Letter]

**Acceptance summary:**

This work shows how changes in butterfly wing pigmentation develop and evolve. More specifically, genetic analysis, analysis of gene expression, and characterization of scale structure was used to identify the likely causal genetic variation in the cortex locus that is responsible for the presence/absence of the yellow band on the wings of some Heliconius butterflies.

**Decision letter after peer review:**

[Editors’ note: the authors submitted for reconsideration following the decision after peer review. What follows is the decision letter after the first round of review.] 

Thank you for submitting your work entitled "The gene cortex controls scale colour identity in Heliconius" for consideration by *eLife*. Your article has been reviewed by 3 peer reviewers, one of whom is a member of our Board of Reviewing Editors, and the evaluation has been overseen by a Senior Editor. Reviewers 1 and 2 have opted to remain anonymous. Review 3 is from a collaborative peer-review group, and some members chose to share their identity publicly, as indicated below.

Our decision has been reached after consultation between two of the reviewers. Based on these discussions and the individual reviews below, we regret to inform you that your work will not be considered further for publication in *eLife*.

While the reviewers and editors feel that the topic of your work is timely and intriguing, the reviewers raise several issues with the study that would be difficult to address within the *eLife* revision window. For example, the reviewers propose that the major conclusions of the manuscript are not supported by the data presented, and that a full set of SEM data across all scale type color transformations should be presented. The reviewers also feel that, given the results presented, the relationship between cortex expression and the actual pigmentation remains unclear, and the sole phenotypic analysis is insufficient to make conclusions about the role of a gene in producing pigmentation pattern variation.

*Reviewer #1:*

This is an interesting but complex study that examines the role of a few genes in a previously mapped interval in being the "switch" gene that regulate the presence or absence of a yellow band in the wings of Heliconius butterflies. The study first examines whether there is a correlation between expression level of several (47) genes in the mapped interval in developing wings (or parts of wings) in two separate species of Heliconius each having a race with the yellow band and a race without the yellow band. This part of the study highlights three genes (among others) that show some pattern of differential regulation but shows that there is no simple correlation between the expression level of these three genes in either larval or pupal wings and the presence of the yellow band. The authors then examine the function of one of the genes in the interval, cortex, in scale color development by using CRISPR. They find that cortex crispant individuals display color changes across the whole wing, not just in the region of the yellow band. In particular the black scales (Type II) become white or yellow (Type I), and the red scales (Type III) also become white or yellow (although this last transformation is not documented at the SEM level). The authors examine, once again, the expression domain of the cortex gene, this time during pupal development with an antibody, and they find that the gene is expressed across the whole wing, supporting its functional effects also across the whole wing. They observe that cortex is expressed in multiple punctate domains in the nuclei of scale building cells, which are polyploid cells, and in a single punctate domain in the nucleus of non-scale building epidermal cells, which are not polyploid. They then test whether perhaps there are more of these punctate nuclei in the region of the yellow band, but they find no such correlation.

In the end the authors try to argue that either (1) cortex is the yellow-band switch gene they are after but that the switch is not in the form of a typical spatially expressed gene (in the shape of the yellow band) but perhaps in the form of some threshold or heterochronic mechanism (not clearly explained), or that (2) another gene in the mapped interval, not examined for function in this study, is instead the switch genes they are after, and which may (or may not) interact with cortex in the differentiation of the yellow band.

I believe the authors are trying hard to implicate cortex in some way, as the yellow band switch locus, but the data just does not support this. Instead the authors implicate cortex in scale color identity (the title of the manuscript). However, given that cortex (alone) cannot control a specific color either, because the effect of cortex on color is different in different parts of the wing, their model for how cortex acts is too simple and does not fit their data. A combinatorial genetic code for both scale color and morphology (see below), where cortex is simply one of the players (rather than a major switch/homeotic gene) is required to explain the data in this manuscript.

Furthermore there are several data missing from the manuscript that need to be added to support some of the conclusions drawn, and several other data that would be important to add for purposes of data replication across labs.

1) The authors claim that cortex converts Type II (black) scales into Type I (white/yellow) scales but their SEM data and scale morphological measurements presented in the supplement don't fully support this conclusion. These transformations vary from species to species (e.g. H. melpomene and H. erato show different degrees of transformation) and only some features of the scale are actually transformed (e.g., cross rib periodicity in both species, and scale width and length and ridge periodicity in H. melpomene). The remainder of the measurements show that cortex is not sufficient to convert scale Type II into scale Type I.

2) I suggest that the definition of the scale types presented should be made more explicit. What are scale types I, II and III really? In line 87 it is mentioned that these scale types are based on scale color and on scale morphology but what follows is just a description of the pigments found in each scale and not their morphology. Furthermore, the data presented in the manuscript suggests that color and morphology can be uncoupled with genetic perturbations of cortex, so, it is even useful to stick to this scale type nomenclature going forward? Something to consider.

3) There is a need for a new figure showing how the scale morphological measurements were actually conducted. There is no scale bar in the SEM images of yellow and black scales and this should be added. The SEM images used to represent a typical yellow WT scale and a transformed yellow scale of H. melpomene (in Figure 7) show very different densities of cross-ribs (but I am not even sure what exactly is being considered a cross-rib), yet the graph indicates that there is no difference between these scale types. This is confusing and needs clarification. Make sure you look up scale morphology nomenclature in Ghiradella 1991 (Applied Optics) to make sure you designate ribs (crossribs) and microribs appropriately. There seem to be quite a lot of differences in microrib density across Wt scales and transformed yellow scales in H. melpomene.

4) The authors claim that cortex converts Type III (red) scales into Type I (white) but they only describe conversions of Type II (black) into Type I (yellow) scales at the SEM level and don't provide any SEM images or quantitative data for the red to yellow, red to white, and black to white scale transformation. Adding these data is important to support the conclusions of the study.

5) I suggest the authors remove the dome-t and dome/washout gene data from the manuscript as 1) nothing about these genes is mentioned in the abstract; 2) the expression of these genes doesn't correlate with presence of the yellow band; 3) the genes are not investigated at the functional level; 4) the whole gene duplication issues surrounding these genes make the whole manuscript more difficult to read and does not, in the end, contribute to the main story that yielded results – which is the function of cortex in scale development. The function of these genes might still be worthy of investigating using CRISPR at a later date, and perhaps it would be useful to include the expression pattern data in that subsequent paper. This is merely a suggestion that I believe will make this manuscript less heavy and easier to read by focusing the reader's attention on the main points of this story.

6) Pigmentation and scale morphology is most likely controlled at the pupal stages of wing development and by measuring RNA levels of candidate switch genes at just two time points during pupal development (36hrs and 60-70 hrs after pupation) you may not have sampled the correct time window for yellow band differentiation. Several genes are expressed only during the first 16-30 hrs of pupal development, in species that need 7 days for pupal development (see Monteiro et al. 2006 for genes such as Wg, pMad and Sal) so sampling wings (for RNA-seq and antibody stains) at 36hrs and 60-70 hours may not be an ideal sampling strategy going forward.

7) The authors mention that because cortex causes changes in both scale color and morphology this suggests "that cortex acts during early stages of scale cell fate specification rather than during the deployment of effector genes". This conclusion needs more discussion. Matsuoka and Monteiro (2018) showed that knockout of the gene yellow, an effector gene at the end of a gene regulatory network for melanin pigment production, also led to both changes in scale color and morphology. These authors proposed instead that absence of certain pigments on the wing, such as dopa melanin, caused chitin to polymerize differently and form an extra lamina that prevent the windows from forming in the scales (just as seen in cortex mutants). The authors should consider and evaluate this alternative explanation in their discussion.

8) Did the authors examine whether there were protein coding changes between the 47 genes in the mapped interval between the yellow and black races? Please mention whether this was done. Please also upload the sequences of the genes that were studied and provide accession numbers for these sequences.

*Reviewer #2:*

This manuscript explores the role of the gene cortex in the specification of wing scales in the butterfly genus Heliconius. Species of Heliconius butterflies are notorious for their reciprocal mimicry of wing color patterns. Several genes are known to control variation of specific color pattern elements within and between species, cortex is one of them. The authors combine RNAseq analysis across wing development, in situ hybridizations, antibody stainings and analysis of crispr somatic mutations to dissect the role of cortex in the specification of scales. Their main claim is that cortex imparts scale identity (color, morphology), namely type II and type III identity.

Although this paper includes a substantial amount of work and a number of interesting observations, I am not sure what can really be concluded in the end, and several results would need follow-up experiments to reach a stable conclusion.

The strongest part, in my opinion, is the analysis of somatic mutant clones of cortex in the wings of different species. The authors show that the lack of cortex consistently results in the conversion of type II and type III scales into type I scales, and thereby demonstrate the necessity of this gene for type II and III identity. This is solid, interesting, but not a novel concept from a genetic or developmental biology point of view. There are countless examples in the 1990s literature of genes whose mutations results in such shifts in cell identity (e.g., poxn and cut in the peripheral nervous system of flies).

From this result, two questions emerge: how and when does cortex assign this identity during development? And how does cortex explain the variation in color pattern among Heliconius morphs and species? Although the paper discusses these two questions, I find the answers unclear and the results confusing.

The authors first examine the expression dynamics of cortex. They re-annotated the 47-gene genomic interval where cortex maps and analyzed the differential expression of all genes in the interval, across developmental stages, across species and morphs and also compared wing compartments.

1. Their main conclusion is that cortex is the most likely candidate in this interval to explain color pattern variation. I am not sure why the authors did this. I thought this was already clearly established from a previous paper (Nadeau et al. 2016, Nature).

2. Moreover, the explanations of the differential gene expression (DGE) analysis are often too shallow to really understand what the authors really did, including the method description. The figures are poorly annotated and it's difficult to understand if there are replicates in the RNA-seq analysis.

3. One striking result from this part, is that the DGE suggests that cortex is differentially expressed in the 5th instar larvae between 2 morphs of Heliconius erato and 2 morphs of Heliconius melpomene, but the differential expression goes into opposite directions between these 2 species. How could the same phenotypic variation between morphs of 2 species be caused by opposite DGE? They authors note that it is interesting but do not comment or analyze further.

4. They pursue their investigation with in situ hybridization on 5th larval instar wings and mitigate the notion of a spatial correlation between cortex transcripts spatial distribution and color patten elements proposed by Nadeau et al., 2016. Here again, the figure would benefit from better annotation. The authors indicate subtle differences in the local distribution of cortex transcripts between morphs but do not really conclude anything from their observation. They also give no indication of sample size or replicates, which I find unsettling given the noise associated with this experiment. I am not sure what this figure really adds to the published work, or to the present manuscript.

5. Finally, the authors examine the distribution of Cortex protein in late (2-day pupa) developing wings with a polyclonal antibody. They find, surprisingly, that the protein is distributed more or less uniformly in the wing epithelium and localizes to the cell nuclei. While this is very different from the patterned transcript distribution, it is consistent with the somatic mutant clone analysis that showed that any mutated cell at any position of the wing displayed a phenotype. But this opens many questions: what is the origin of the apparent difference in expression between protein and transcripts? Is cortex secreted and it diffuses across the wing? Or is the transcript expression spatially dynamic and the protein distribution revealed by the authors reflects the temporal integration of this expression? And if Cortex is present and functional across the wing, how does it produce discrete pattern elements?

The authors conclude their paper with a figure suggesting that cortex specifies typeII/III scale identity early during wing disc development and that the distinction between type II and type III is subsequently governed by the gene optix at a later stage. But what substantiates the idea that cortex imparts cell type identity early on? What does Cortex larval (5th instar) distribution look like? Is it as uniform as that of later stages? The data presented here do not offer the temporal or functional resolution to support this conclusion.

In conclusion, this paper shows that the mutation of the gene cortex results in scale type transformation, but fails to explain or suggest how this may happen during development. It also does not suggest how cortex may control the "fantastically diverse" pattern variation in Heliconius. I find this study insufficient to justify publication in *eLife*.

*Reviewer #*3:PREreview of "The gene cortex controls scale colour identity in Heliconius"

Authored by Luca Livraghi et al. and posted on bioRxiv DOI: 10.1101/2020.05.26.116533

Review authors in alphabetical order of last name:

Monica Granados, Vinodh Ilangovan ORCiD (http://orcid.org/0000-0002-3445-5383), Katrina Murphy, Aaron Pomerantz

This review is the result of a virtual, live-streamed preprint journal club organized and hosted by PREreview and eLife. The discussion was joined by 17 people in total, including researchers from several regions of the world.

Overview and take-home message: In this preprint, Livraghi et al. present noteworthy advances in evolutionary biology by characterizing the role of cortex gene in multiple Heliconius butterfly species, which is responsible for the wing patterns: yellow bar or the Type I scale cell fates (white/yellow). The authors identified cortex gene's major role in sympatric speciation, the modulation of convergent wing patterns, and the regulation of scale identity in multiple Heliconius species, which naturally have different niches to help explain different co-mimetic morphology. Livragi's team provides strong evidence for the cortex gene as one of the earliest regulators and its ability to set the differentiation of scale cells in a molecular switch fashion from yellow to red/black at a particular development stage through distal localization. This important discovery on the role of cortex gene fills a gap in our existing knowledge about the gene's ability to control scale cell identity and wing color patterns. Since this work is of significant interest in evolutionary biology, we outlined some concerns below that could be addressed in the next version.

Positive feedback:

* We strongly recommend this preprint to others/for peer review. In addition, we recommend this article to trainees as educational material to learn evolutionary developmental biology through interactive tutorials.

* The authors have provided a good amount of novel results and have utilized current tools to address their questions.

* This research fills a gap in our understanding of wing patterning in Heliconius while doing so in a very comprehensive way across multiple species and using techniques that systematically detail the association between gene expression and phenotype.

* It was interesting to learn that the cortex gene doesn't follow the typical pattern gene paradigm. We do not have many examples of integrator genes like cortex, which give binary outputs from a network of genes and integrate elements to produce a singular output.

* This is a textbook example and is important for evolutionary development and mimicry studies. It is hard to find and/or work with a developmentally important gene that is amenable for genetic modification and still be able to work with viable offspring and have it be relevant for evolution.

The current cortex protein data as seen in Figure 6 adds novel data to the manuscript.

Thanks to the authors for setting a great example of showing modeling information. The graphics are visually appealing and convey complex information well.

* This preprint sets up a good next step of how cortex evolved in a more broad context. We know the cortex gene is potentially implicated in wing pattern evolution in other distantly related butterflies and moths (e.g. peppered moth Biston betularia) and in possible roles of evolution/speciation by pattern changes due to genomic inversions at cortex locus.

* The authors did a good job of creating a well-composed manuscript. Yellow bar with one species had a contradiction but did reconcile with further research questions.

* Definitely, [the results are likely to lead to future research] especially with understanding how a cell cycle regulator affects developmental cell fates in terms of these scale colors and structures.

* Antibodies can open up future research. This research team figured out three elements and there are possibly more to explore. Future research might investigate how cortex possibly regulates endocycling and what this means for color identity determination.

Major concerns:

1. The use of the term "race" to define butterflies with specific phenotypes needs to be revised to clines or strains or variants. "Race" is a social construct and not a biological reality and we strongly suggest revising this term.

2. The authors state that cortex and dome/wash genes are controlled by inversion (see Line 375, page 19). Does the strain they engineered have/carry the inversion?

– We are aware that inversion for species is complex – strains, genetic background – starting material for inversion.

– Inversion events occurred millions of years ago in the loci contributing to the wing pattern. Authors describe the first generation of CRIPSR knock-outs in Heliconius sp. and hence we suggest to include further information.

3. We strongly suggest the authors elaborate on their qRT-PCR analysis pipeline. Did the authors follow MIQE guidelines (https://academic.oup.com/clinchem/article/55/4/611/5631762) in their quantitative real time PCR assays?

4. More explanation could be provided for cortex protein experiments. Figure 6 could explicitly say what developmental stage/time after pupation (they report this in the Methods section) and the rationale behind presenting data for this stage in development.

– Maybe perform a systematic developmental time series of cortex immunostaining experiments?

5. We recommend the authors mention institutional or local animal care ethical approval and safety regulations in the field working on Heliconius sp. for setting best practice reporting standards.

6. We suggest to clarify the lack of a clear correlation between in situ stains and the mutational effects of cortex CRISPR knock-outs.

7. Could a sized-down Figure S10 be added to Figure 6 in the manuscript to provide more information about the nuclear ploidy and cortex antibody signal? Even no association is informative and helps the reader think about the connection between color/endopolyploidy.

Acknowledgments:

We thank all participants for attending this preprint journal club. We especially thank those that engaged in the discussion. Their participation contributed to both a constructive and lively discussion.

Below are the names of participants who wanted to be recognized publicly for their contribution to the discussion:

Monica Granados | PREreview | Leadership Team | Ottawa, ON

Vinodh Ilangovan | Labdemic - Founder |Postdoc | @I_Vinodh

Katrina Murphy | PREreview | Project Manager | Portland, OR

Aaron Pomerantz | UC Berkeley/Marine Biological Laboratory | Ph.D. Candidate | Berkeley, CA/Woods Hole, MA

[Editors’ note: further revisions were suggested prior to acceptance, as described below.]

Thank you for submitting your article "Cortex cis-regulatory switches establish scale colour identity and pattern diversity in Heliconius" for consideration by *eLife*. Your article has been reviewed by 3 peer reviewers, and the evaluation has been overseen by Patricia Wittkopp as the Senior and Reviewing Editor. The following individual involved in review of your submission has agreed to reveal their identity: Antonia Montiero (Reviewer #1).

Essential revisions:

1) In line 350 the authors mention that the presence of an upper lamina is an important morphological feature of yellow/white scales and then they cite Matsuoka and Monteiro (2018). This paper shows, instead, that the white lamina forms in scales that are brown in color, not yellow nor white. In addition, several papers have shown the presence of this type of upper lamina in silver colored scales. While the presence of this lamina may indeed be a feature of Type I scales in Heliconius butterflies, I would refrain from attaching too much importance to this lamina regarding the formation of a particular color. The color changes observed in Heliconius butterflies are most likely caused by changes in pigmentation than by changes in the presence/absence of this lamina.

2) Line 160: Spell out what APC/C motifs are.

3) In Figure 9e I cannot really see/understand the effect of cortex disruptions on red scale phenotypes – both the SEM image provided and the low-resolution image of the red-colored scale are low quality. Please provide higher quality images for these data. In particular, the SEM data does not show a scale type III converting into a scale type I.

4) Line 375: Why "key early scale cell specification switch"? What does "early" refer to and what data indicates time of gene activity in the manuscript?

5) Line 376: How does the sentence above lead to the next sentence "and thus has the potential to generate much broader pattern variation than previously described patterning genes"? I don't follow what the authors are trying to convey here.

6) Lines 447-449 need revision.

7) Crossribs are sometimes written as cross-ribs. Be consistent.

8) Ln 85 Capitalize van Belleghem.

9) Lns 92-99. The authors made a marked improvement in detailing the morphological differences in the text. The authors also provide a nice visual summary of the pigment of morphological differences in Figure 9f. However, the readers suffer by not being able to connect the text in the introduction to the illustrations in Figure 9f. I understand the challenge is that figures must be referenced in order, and it definitely makes sense for Figure 9f to be placed with the CRISPR images. If possible, I would urge the journal to allow the authors to add "(see Figure 9f)" in lines 92-99, as I think it would help the reader and retain the author's figure order. The alternative would be including a version of the cartoon in 9F in Figure 1, which could be considered since the scale structure and fate are such a major component of the study.

10) Lns 130-137. I agree with Reviewer 1 that the dome/wash results detract from the main points and flow of the paper. However, I don't agree that they should be removed. Rather, I recommend that the detailed description of the dome duplication be moved to supplementary materials, as it seems the primary purpose of including the duplication description is to justify analyses were performed and interpreted accurately.

11) Ln 237. The authors state "the sharp boundaries observed between wild-type and mutant scales suggest cortex functions in a cell-autonomous manner, with little or no communication between neighbouring cells", however, I'm not convinced of this. It seems the mutant scales of single individuals are more often localized in wing regions, often with neighboring clusters of mutant scales. I understand that the sharp boundaries may suggest little cell communication, later in wing pattern development. However, the localization of clusters suggests cortex may impact communication between cells in early wing cell replication. This doesn't have much of an impact on their findings, but the authors may want to consider tempering the interpretation.

12) Ln 352-353. The authors state "…cortex is necessary for the development of lamellar structures in Heliconius scales.". This statement is too broad and encompassing. Figure 9 shows that cortex appears to be required to develop Type II and III scales with open upper lamina. In this regard, cortex is not required for the development of lamellar structures but rather required to regulate the development of architectural variation in lamellar structures. This may seem minor, but as worded it reads that "cortex is necessary for the development of lamellar structures", which I don't agree is supported by the data.

13) Ln 393-398. I find the authors attempt to explain the difference in the direction of cortex DGE between species to be quite brief and hand-waving. I'm not clear what is meant by "some relatively subtle developmental heterochrony between the two species would capture the state of differentially expressed genes in a different dynamic step." It seems the timing and levels of cortex expression could be quite precise and variable over short periods of time. In such cases, the limited sampling in each species could simply be detecting different points. In other words, it may be more informative for the reader to have the authors state that the current resolution may simply be insufficient to resolve and compare the precise cellular functions of cortex in the two species. And, that the difference between species suggests more detailed examination, perhaps even real-time expression data of cortex, may be needed.

14) Figure 10. I would consider including WntA in Figure 10. The timing of expression of WntA effecting melanic pattern is well resolved (5th instar), the authors discuss how WntA and cortex may interact, so it would seem reasonable to include WntA in Figure 10, so that the figure offered the best present model for the timing and placement of each major color patterning gene during wing development.

15) Ln 510. Italicize H. melpomene and H. erato.

16) Ln 684. Ray et al. 2019 not included in References.

17) Ln 885-886. McMillan et al. 2020 reference is incomplete.

18) Figure S15. Please remove "Lorem ipsum" from the "H. erato ridge periodicity" plot.

19) The introduction lacks a short description of the concrete wing colour patterns of the studied Heliconius species. It would be helpful for some readers to have this information between lines 100-106: Colour stripes, location on the wing, colour polymorphism,…

20) I agree with a previous reviewer that the data on cortex and dome/wash differential gene expression could be removed from this manuscript (although it should be published elsewhere). However, in the current form the manuscript starts with results that are difficult to interpret and this is confusing. The rational to investigate cortex function is already given by Nadeau et al. 2016.

21) The manuscript could therefore gain clarity if the authors re-arrange their very interesting findings. 1) CRISPR mosaics to characterize cortex function and SEM analysis on scale type 2) Identification and characterization of the CRE's and the role in stripe determination 3) cortex expression analysis that does not seem to associate with a hindwing stripe.

22) Mosaic loss of function of cortex apparently reveals a loss of melanic (Type II) and red (Type III) scales. The authors note sharp boundaries between wild-type and mKO scales and suggest cortex functions to be cell-autonomous with little or no communication between neighbouring cells. However, in Figure 4b and S10 they also report intermediate white/red scale phenotypes in scales of the "red band element". These scales sometimes have a patchy/granular red colouration. The authors further note an asymmetric deposition of pigment across the scale but should better point out what this means. Is the result conform with / contradicting their interpretation of a cortex mediated scale identity switch? Could the red pigment presence have non-autonomous sources from the surrounding tissue?

23) The Figures are very nice, but many legends lack a certain degree of precision in the description of each panel. The authors need to revise this with care, this has already been pointed out by a previous reviewer. Please indicate what the illustrations literally present. For example, Figure 1a, horizontal black bars are coding sequences? gene predictions? mapped transcripts? other elements? What reference genome assembly? Do they indicate the 47 candidate genes or just a small portion of it? Wing patterns are present on the dorsal or ventral side of the wing? Figure 1B: Are yellow and blue dots reported sampling locations?

24) Conclusion: The first part (lines 492 – 505) appears to be redundant with lines 511 -514 in the second paragraph. Maybe integrate the first paragraph into the discussion.

*Reviewer #1:*

In this paper the authors associate genetic variation in regulatory sequences of the gene cortex with the presence/absence of a yellow band of color in the wings of two species of Heliconius butterflies. They show that cortex is spatially regulated in larval wings, but the expression of this gene does not correlate with the presence or absence of the yellow band. Then they show that the gene is expressed in the nuclei of all cells of the pupal wing. By disrupting cortex they show that black cells (Type II) become white or yellow (Type I), and red scales (Type III) become paler across the whole wing.

By examining open regions of chromatin around cortex, they discover that at least in one of the species, the insertion of two transposable elements in an open region of chromatin associates with the presence of the yellow band. They show that disrupting this regulatory region in a race of butterflies that does not contain the yellow band, nor the TE insertions, leads to the loss of the black color in a band-like shape, and the appearance of yellow scales in that region of the wing. They identify a different region of open-chromatin in the other Heliconius species that when disrupted also leads to the transformation of black scales into yellow scales in a band-like pattern.

The authors achieved their aims and the results support their conclusions.

The strength of this manuscript lies in the use of multiple approaches to identify the likely causal genetic variation in the cortex locus that is responsible for the presence/absence of the yellow band. The only weakness (if I can call it that) is that it is still not clear how cortex, which is also expressed in the nuclei of the yellow scales in races that supposedly have the TE insertion and closed chromatin in that enhancer region, fail to develop black scales in that region of the wing.

This is one of the first few papers that examines the function of specific open regions of chromatin in the DNA of butterfly species using CRISPR-Cas9. The main novelty of this paper is in identifying how a gene with a homogeneous expression pattern across the wing (during the pupal stage) can still have "hidden" modular regulatory regions that drive unique functions (albeit not expression) is specific regions of the wing.

This work reminds me of the regulation of the vestigial gene in the wings of *Drosophila*. vestigial also has homogeneous expression across the wing pouch but it achieves this homogeneous expression via two separate enhancers that have complementary expression patterns.

*Reviewer #3:*

The gene cortex was reported to control mimicry and crypsis in butterflies (Nadeau et al. 2016). This study finds cortex function to be essential for Heliconius wing scale type determination at the transition from scale type I to type II / III. This is shown by genetic loss of function assays and characterization of scale structure by scanning electron microscopy. In particular, the authors show that cortex function is essential for scale type determination throughout the wings that mainly contain type II/III scales in Heliconius butterflies. This is revealed by a series of CRISPR/Cas9 derived somatic mosaic mutants in diverse genetic backgrounds and species. Expression of a specific yellow (type I scale) hind-wing stripe in some Heliconius melpomene and H. erato morphs was found to depend on molecular tinkering and malfunction of a discrete cortex cis-regulatory element (CRE). The authors identify distinct CRE's in both species by ATAC-seq open chromating mapping and narrow down candidate regions by genetic association to the yellow stripe. Hi-C assays were used to verify that the elements indeed interact with the cortex promoter. However, a possible regulation of other genes cannot be excluded. Tinkering of these elements appears to be a natural mechanism in wing colour pattern evolution, since a yellow stripe morph is associated with an insertion of a transposable element in the corresponding region in the morph H. melpomene timareta. Expression of cortex was investigated at different developmental stages by in-situ hybridization and immuno-staining techniques. Cortex transcripts reveal complex expression pattern that do not seem to be associated with the yellow hindwing stripe in corresponding morphs. Cortex protein is localized in the cell nucleus throughout wing cells and future studies must resolve how cortex regulatory elements determine such specific stripe-pattern. This article contrasts the widespread expression of cortex with a complex transcriptional regulation of this gene and scale type transition in discrete wing domains. The authors argue that cortex is a prime target for wing pattern evolution, acting as "input-output" module, whereby complex spatio-temporal information is translated to determine scale type and colour.

**Author response:**

[Editors’ note: the authors resubmitted a revised version of the paper for consideration. What follows is the authors’ response to the first round of review.]

*Reviewer #1:*

[…] In the end the authors try to argue that either (1) cortex is the yellow-band switch gene they are after but that the switch is not in the form of a typical spatially expressed gene (in the shape of the yellow band) but perhaps in the form of some threshold or heterochronic mechanism (not clearly explained), or that (2) another gene in the mapped interval, not examined for function in this study, is instead the switch genes they are after, and which may (or may not) interact with cortex in the differentiation of the yellow band.

I believe the authors are trying hard to implicate cortex in some way, as the yellow band switch locus, but the data just does not support this. Instead the authors implicate cortex in scale color identity (the title of the manuscript). However, given that cortex (alone) cannot control a specific color either, because the effect of cortex on color is different in different parts of the wing, their model for how cortex acts is too simple and does not fit their data. A combinatorial genetic code for both scale color and morphology (see below), where cortex is simply one of the players (rather than a major switch/homeotic gene) is required to explain the data in this manuscript.

Furthermore there are several data missing from the manuscript that need to be added to support some of the conclusions drawn, and several other data that would be important to add for purposes of data replication across labs.

1) The authors claim that cortex converts Type II (black) scales into Type I (white/yellow) scales but their SEM data and scale morphological measurements presented in the supplement don't fully support this conclusion. These transformations vary from species to species (e.g. H. melpomene and H. erato show different degrees of transformation) and only some features of the scale are actually transformed (e.g., cross rib periodicity in both species, and scale width and length and ridge periodicity in H. melpomene). The remainder of the measurements show that cortex is not sufficient to convert scale Type II into scale Type I.

Thank you for bringing this to our attention. We agree that the data does not clearly show complete homeosis. We therefore have softened our language, arguing for incomplete homeosis, but show that major structural rearrangements do accompany cortex functional perturbations. However, our cortex KO scales do show the presence of a lamina covering the scale windows as well as the presence of microribs joining the longitudinal ridges, features unique to otherwise wild-type Type I scaleswhich always occur in wild type and mutant Type I scales and never occur in type II or type III scales. We believe that nanomophometric measurements might not reflect full homeosis not because of a lack of transformation, but because of small positional differences known to affect scale structure across the wing (see Day et al., 2019). We believe the qualitative nature of the SEM images clearly reflect these structural changes, which are at least sufficient to show cortex perturbations affect both pigmentation state and ultrastructural features in a consistent way. We have also uploaded all SEM images, which include a large number of scales that consistently show homeosis in microribs and lamina, to the Dryad data repository.

2) I suggest that the definition of the scale types presented should be made more explicit. What are scale types I, II and III really? In line 87 it is mentioned that these scale types are based on scale color and on scale morphology but what follows is just a description of the pigments found in each scale and not their morphology. Furthermore, the data presented in the manuscript suggests that color and morphology can be uncoupled with genetic perturbations of cortex, so, it is even useful to stick to this scale type nomenclature going forward? Something to consider.

We agree. We have expanded upon these definitions in the introduction (See lines 89-99). We also added the citation of Janssen, Monteiro and Brakefield 2001, which argues for a strong coupling between pigmentation and ultrastructure based on wounding experiments.

3) There is a need for a new figure showing how the scale morphological measurements were actually conducted. There is no scale bar in the SEM images of yellow and black scales and this should be added. The SEM images used to represent a typical yellow WT scale and a transformed yellow scale of H. melpomene (in Figure 7) show very different densities of cross-ribs (but I am not even sure what exactly is being considered a cross-rib), yet the graph indicates that there is no difference between these scale types. This is confusing and needs clarification. Make sure you look up scale morphology nomenclature in Ghiradella 1991 (Applied Optics) to make sure you designate ribs (crossribs) and microribs appropriately. There seem to be quite a lot of differences in microrib density across Wt scales and transformed yellow scales in H. melpomene.

We apologise for the confusion in nomenclature. We have since clarified the structures as well as updated the SEM figure to reflect this (See Figure 9, f). The precise methodology relating to the scale measurements can be found in Day et al., 2019. We believe this should be enough to locate details on measurement reproducibility, but are happy to include a reproduction of the figure from Day et al. 2019 if the reviewers request this.

4) The authors claim that cortex converts Type III (red) scales into Type I (white) but they only describe conversions of Type II (black) into Type I (yellow) scales at the SEM level and don't provide any SEM images or quantitative data for the red to yellow, red to white, and black to white scale transformation. Adding these data is important to support the conclusions of the study.

We agree and have added the requested images and measurements to the new manuscript (see Figure 9 and Supplementary File 15).

5) I suggest the authors remove the dome-t and dome/washout gene data from the manuscript as 1) nothing about these genes is mentioned in the abstract; 2) the expression of these genes doesn't correlate with presence of the yellow band; 3) the genes are not investigated at the functional level; 4) the whole gene duplication issues surrounding these genes make the whole manuscript more difficult to read and does not, in the end, contribute to the main story that yielded results – which is the function of cortex in scale development. The function of these genes might still be worthy of investigating using CRISPR at a later date, and perhaps it would be useful to include the expression pattern data in that subsequent paper. This is merely a suggestion that I believe will make this manuscript less heavy and easier to read by focusing the reader's attention on the main points of this story.

We thank the reviewer for the suggestion. However, we believe it is useful to point out there are multiple genes at the locus that show patterns of differential expression, especially as these genes have been implicated in pattern evolution in other studies and might be useful for future studies to follow up on.

6) Pigmentation and scale morphology is most likely controlled at the pupal stages of wing development and by measuring RNA levels of candidate switch genes at just two time points during pupal development (36hrs and 60-70 hrs after pupation) you may not have sampled the correct time window for yellow band differentiation. Several genes are expressed only during the first 16-30 hrs of pupal development, in species that need 7 days for pupal development (see Monteiro et al. 2006 for genes such as Wg, pMad and Sal) so sampling wings (for RNA-seq and antibody stains) at 36hrs and 60-70 hours may not be an ideal sampling strategy going forward.

This is an important point. While we agree that we have likely not captured the terminal differentiation factors, we believe our new data clearly shows that cortex is a key factor that establishes the identity of Type II and III scales early in development. Furthermore, population genetic scans show that regions around the cortex gene are the only ones that are differentiated between populations differing in the presence of the yellow band, suggesting this region must be the causative locus, and not terminal pigmentation genes or other transcription factors.

7) The authors mention that because cortex causes changes in both scale color and morphology this suggests "that cortex acts during early stages of scale cell fate specification rather than during the deployment of effector genes". This conclusion needs more discussion. Matsuoka and Monteiro (2018) showed that knockout of the gene yellow, an effector gene at the end of a gene regulatory network for melanin pigment production, also led to both changes in scale color and morphology. These authors proposed instead that absence of certain pigments on the wing, such as dopa melanin, caused chitin to polymerize differently and form an extra lamina that prevent the windows from forming in the scales (just as seen in cortex mutants). The authors should consider and evaluate this alternative explanation in their discussion.

We thank the reviewer for this important comment. The idea that cortex is likely acting at an early stage came mainly from the fact that this is when we see differential expression. However, given that we see cortex protein present until at least 80hr post pupation formation, we agree that it is possible for cortex to be acting later on in development too. We have thus removed this sentence.

8) Did the authors examine whether there were protein coding changes between the 47 genes in the mapped interval between the yellow and black races? Please mention whether this was done. Please also upload the sequences of the genes that were studied and provide accession numbers for these sequences.

We did not check across all the genes in the interval, however, Nadeau et al. (2016) did show there was no evidence of fixed protein coding changes at cortex itself, and that other genes in the Yb locus as defined in that paper were not associated with SNPs (see extended table 1 of that paper). Note that the genes cortex and wash were the only genes in the locus with any associated SNPs. We can add repeat this analysis if the reviewer deems this necessary. The genes are available as annotations from the H. erato demophoon and H. melpomene melpomene genomes (found on lepbase.org). Supplementary File 4 contains all the genes within the interval as well as their corresponding Gene ID for both H. erato and H. melpomene genomes. Hopefully this is sufficient, we are happy to further upload individual sequences to ENA if this is necessary.

Reviewer #2:

[…] Although this paper includes a substantial amount of work and a number of interesting observations, I am not sure what can really be concluded in the end, and several results would need follow-up experiments to reach a stable conclusion.

The strongest part, in my opinion, is the analysis of somatic mutant clones of cortex in the wings of different species. The authors show that the lack of cortex consistently results in the conversion of type II and type III scales into type I scales, and thereby demonstrate the necessity of this gene for type II and III identity. This is solid, interesting, but not a novel concept from a genetic or developmental biology point of view. There are countless examples in the 1990s literature of genes whose mutations results in such shifts in cell identity (e.g., poxn and cut in the peripheral nervous system of flies).

From this result, two questions emerge: how and when does cortex assign this identity during development? And how does cortex explain the variation in color pattern among Heliconius morphs and species? Although the paper discusses these two questions, I find the answers unclear and the results confusing.

The authors first examine the expression dynamics of cortex. They re-annotated the 47-gene genomic interval where cortex maps and analyzed the differential expression of all genes in the interval, across developmental stages, across species and morphs and also compared wing compartments.

1. Their main conclusion is that cortex is the most likely candidate in this interval to explain color pattern variation. I am not sure why the authors did this. I thought this was already clearly established from a previous paper (Nadeau et al. 2016, Nature).

Thank you for the comment. We agree that Nadeau et al. show compelling evidence for the involvement of cortex in establishing pattern differences, however, their conclusions were drawn from microarray tiling experiments examining differences between H. melpomene morphs that did not differ specifically in the presence of a yellow bar (H.m.plesseni and H.m.malletti). We believed it necessary to expand on these analyses by showing cortex was also differentially expressed in association with the yellow bar phenotype, which is the phenotype of focus in our manuscript.

2. Moreover, the explanations of the differential gene expression (DGE) analysis are often too shallow to really understand what the authors really did, including the method description. The figures are poorly annotated and it's difficult to understand if there are replicates in the RNA-seq analysis.

We apologise for the lack of clarity regarding the differential expression experiment. We have updated the methods and expanded upon the analysis, including further analysis in Supplementary File 4. We hope this is now clearer to follow.

3. One striking result from this part, is that the DGE suggests that cortex is differentially expressed in the 5th instar larvae between 2 morphs of Heliconius erato and 2 morphs of Heliconius melpomene, but the differential expression goes into opposite directions between these 2 species. How could the same phenotypic variation between morphs of 2 species be caused by opposite DGE? They authors note that it is interesting but do not comment or analyze further.

We agree this counter-intuitive results needs more explanation. We have expanded on this in the discussion (See lines 393 to 398). We hope this provides some clarity to the discussion.

4. They pursue their investigation with in situ hybridization on 5th larval instar wings and mitigate the notion of a spatial correlation between cortex transcripts spatial distribution and color patten elements proposed by Nadeau et al., 2016. Here again, the figure would benefit from better annotation. The authors indicate subtle differences in the local distribution of cortex transcripts between morphs but do not really conclude anything from their observation. They also give no indication of sample size or replicates, which I find unsettling given the noise associated with this experiment. I am not sure what this figure really adds to the published work, or to the present manuscript.

Apologies for the lack of clarity regarding the figure. We have employed the use of landmarks using the wing veins to illustrate the differences in expression between different wings/morphs. We have added a supplementary file (Supplementary File 6), showing more replicates across the different morphs. While it is difficult to interpret the results, we believe this is useful information for future researchers wishing to address a possible function for cortex. It seems likely that whatever the mechanism by which cortex is creating these phenotypic differences, early differences in expression are likely crucial. We therefore think this would be a good reference point for future studies to expand upon.

5. Finally, the authors examine the distribution of Cortex protein in late (2-day pupa) developing wings with a polyclonal antibody. They find, surprisingly, that the protein is distributed more or less uniformly in the wing epithelium and localizes to the cell nuclei. While this is very different from the patterned transcript distribution, it is consistent with the somatic mutant clone analysis that showed that any mutated cell at any position of the wing displayed a phenotype. But this opens many questions: what is the origin of the apparent difference in expression between protein and transcripts? Is cortex secreted and it diffuses across the wing? Or is the transcript expression spatially dynamic and the protein distribution revealed by the authors reflects the temporal integration of this expression? And if Cortex is present and functional across the wing, how does it produce discrete pattern elements?

We have expanded upon this analysis to include further time points as well as show that protein localisation matches the in situ expression in 5th instar larvae. We believe the pupal expression are an important result, as these can explain the wing wide effects seen for the CRISPR KOs. We agree with the reviewer, who poses many important, yet unresolved questions. We have tried to address these in the discussion (see lines 378-398 and 407-420), but we believe a more mechanistic deep-dive into the functional role of cortex falls beyond the scope of this manuscript and will be more appropriate for a future study, as our focus deals more with the evolution of the phenotypic switches.

Reviewer #3:

[…] Major concerns:

1. The use of the term "race" to define butterflies with specific phenotypes needs to be revised to clines or strains or variants. "Race" is a social construct and not a biological reality and we strongly suggest revising this term.

Thank you for the comment. We agree and have changed to the use of morph throughout the manuscript.

2. The authors state that cortex and dome/wash genes are controlled by inversion (see Line 375, page 19). Does the strain they engineered have/carry the inversion?

– We are aware that inversion for species is complex – strains, genetic background – starting material for inversion.

– Inversion events occurred millions of years ago in the loci contributing to the wing pattern. Authors describe the first generation of CRIPSR knock-outs in Heliconius sp. and hence we suggest to include further information.

Apologies for the lack of clarity on this. The inversion is present only in the polymorphic H. numata, which appears to have locked these genes into a supergene structure. There is no evidence of a supergene in H. melpomene or H. erato.

3. We strongly suggest the authors elaborate on their qRT-PCR analysis pipeline. Did the authors follow MIQE guidelines (https://academic.oup.com/clinchem/article/55/4/611/5631762) in their quantitative real time PCR assays?

Apologies for the confusion but we are not sure which qRT-PCR experiments the reviewers are referring to, as there was no qPCR experiment in the previous version of the manuscript. We have since added qPCR validation of the RNA-Seq data, and we hope the methodology is adequate in the revised version.

4. More explanation could be provided for cortex protein experiments. Figure 6 could explicitly say what developmental stage/time after pupation (they report this in the Methods section) and the rationale behind presenting data for this stage in development.

– Maybe perform a systematic developmental time series of cortex immunostaining experiments?

Thank you for the comment. We have expanded upon this analysis and have made the staging more explicit, including control experiments in the supplementary files.

5. We recommend the authors mention institutional or local animal care ethical approval and safety regulations in the field working on Heliconius sp. for setting best practice reporting standards.

This is an important point, which other reviewers have also raised. We have added further discussion of these discrepancies (See reviewer 2 comment #4).

6. We suggest to clarify the lack of a clear correlation between in situ stains and the mutational effects of cortex CRISPR knock-outs.

Apologies for the lack of clarity. This was also raised by a previous reviewer and we have expanded upon the methodology used to generate the differential expression statistics. We hope this will be easier to follow.

7. Could a sized-down Figure S10 be added to Figure 6 in the manuscript to provide more information about the nuclear ploidy and cortex antibody signal? Even no association is informative and helps the reader think about the connection between color/endopolyploidy.

We have much expanded upon this analysis and included further stages, as well as control experiments in the supplementary. We hope this will improve clarity.

[Editors’ note: what follows is the authors’ response to the second round of review.]

Essential revisions:

1) In line 350 the authors mention that the presence of an upper lamina is an important morphological feature of yellow/white scales and then they cite Matsuoka and Monteiro (2018). This paper shows, instead, that the white lamina forms in scales that are brown in color, not yellow nor white. In addition, several papers have shown the presence of this type of upper lamina in silver colored scales. While the presence of this lamina may indeed be a feature of Type I scales in Heliconius butterflies, I would refrain from attaching too much importance to this lamina regarding the formation of a particular color. The color changes observed in Heliconius butterflies are most likely caused by changes in pigmentation than by changes in the presence/absence of this lamina.

We thank the reviewer for bringing this to our attention. We have removed the citation and limited our comments to pointing out that *cortex* KOs are accompanied by both ultrastructural differences and pigmentation shifts. See lines 355-359 of the revised manuscript.

*2) Line 160: Spell out what APC/C motifs are.*

Added to line 168.

*3) In Figure 9e I cannot really see/understand the effect of cortex disruptions on red scale phenotypes – both the SEM image provided and the low-resolution image of the red-colored scale are low quality. Please provide higher quality images for these data. In particular, the SEM data does not show a scale type III converting into a scale type I.*

We apologise for the lack of clarity regarding the Type III to I transformation. We have added a supplementary figure that accompanies the Figure 9e that illustrates these changes (Figure 9—figure supplement 2). We note that the transformations to Type I scales are not always complete; some scales appear to be in an “intermediate” state between Type III and I. In the discussion we thus make the argument that *cortex* appears to be necessary for induction of Type III scales, but not sufficient for their proper development (see lines 480-481). Following this logic, it appears that *optix* therefore requires an input from *cortex* to lead to the proper development of red scales. This epistatic interaction has also been shown to occur between certain *H. melpomene* crosses, where the red forewing band shape is dependent on specific *cortex* alleles segregating in the crosses. With regards to the SEMs not showing a full transformation, we note that it was difficult to image and measure the transformed Type III scales, as these immediately curled upon removal for imaging (suggesting that their structural integrity is compromised by *cortex* KOs). However, even in this curled up state, the presence of microribs and a lamina covering the spaces between the microribs and ridges can be seen (we hope this is more evident in the extra images provided in the Figure 9—figure supplement 2). It should also be noted the granular red pigment is internal to the scale and not visible in the SEMs. We believe these data fit our interpretation given in the discussion, where “…*cortex* expression is required for either downstream signalling to *optix*, or to induce a permissive scale morphology for the synthesis and deposition of red pigment in future scales.” A possible mechanism for this observation could perhaps result from differential rates of scale development induced by *cortex* (heterochrony hypothesis). If *cortex* is slowing down scale development, it may allow the scales to enter the timing of *optix* expression and red pigment deposition at an earlier developmental time. In the absence of *cortex*, scales develop quicker (as Type I scales do, see Aymone et al., 2013), and therefore the deposition of red pigment occurs at a time when scales may already be hardening, resulting in this intermediate effect in some scales.

*4) Line 375: Why "key early scale cell specification switch"? What does "early" refer to and what data indicates time of gene activity in the manuscript?*

Thank you for bringing this point to our attention. By “early” we were referring to the fact that differential expression of *cortex* occurs during fifth instar wing development, and so suggests that this is a pre-patterning effect that occurs early during wing development. However, we agree that this term is relative to developmental time chosen to be referred to as early or not, and so we have removed the word. See line 380.

*5) Line 376: How does the sentence above lead to the next sentence "and thus has the potential to generate much broader pattern variation than previously described patterning genes"? I don't follow what the authors are trying to convey here.*

Apologies for the lack of clarity regarding this statement. We were attempting to convey the idea that, in contrast to the other major patterning genes, *cortex* appears to be expressed and affect scale development throughout the entire wing surface, and therefore tinkering with its expression can lead to broad effects throughout the wing. Therefore, rather than it being a canonical “patterning” gene, perhaps it is best conceived as a conserved scale development gene, where it can affect individual scale development trajectories across the wing.

*6) Lines 447-449 need revision.*

Apologies for the confusing phrasing, we have edited this in a way we hope is clearer to the reader. See lines 456-460.

*7) Crossribs are sometimes written as cross-ribs. Be consistent.*

Thank you for alerting us to this. We have corrected it to crossribs throughout.

*8) Ln 85 Capitalize van Belleghem.*

Thank you for spotting this. This has been changed.

*9) Lns 92-99. The authors made a marked improvement in detailing the morphological differences in the text. The authors also provide a nice visual summary of the pigment of morphological differences in Figure 9f. However, the readers suffer by not being able to connect the text in the introduction to the illustrations in Figure 9f. I understand the challenge is that figures must be referenced in order, and it definitely makes sense for Figure 9f to be placed with the CRISPR images. If possible, I would urge the journal to allow the authors to add "(see Figure 9f)" in lines 92-99, as I think it would help the reader and retain the author's figure order. The alternative would be including a version of the cartoon in 9F in Figure 1, which could be considered since the scale structure and fate are such a major component of the study.*

On suggestion by the reviewer, we have added "(see Figure 9f)" to line 92. We hope the journal is OK with this suggestion, otherwise we are happy to reproduce the figure in Figure 1.

*10) Lns 130-137. I agree with Reviewer 1 that the dome/wash results detract from the main points and flow of the paper. However, I don't agree that they should be removed. Rather, I recommend that the detailed description of the dome duplication be moved to supplementary materials, as it seems the primary purpose of including the duplication description is to justify analyses were performed and interpreted accurately.*

Thank you for the suggestion. We have moved the analysis of the duplication to the supplementary. We have retained lines 134-136 however, as these are necessary to interpret the annotations of the gene names in Figure 3.

*11) Ln 237. The authors state "the sharp boundaries observed between wild-type and mutant scales suggest cortex functions in a cell-autonomous manner, with little or no communication between neighbouring cells", however, I'm not convinced of this. It seems the mutant scales of single individuals are more often localized in wing regions, often with neighboring clusters of mutant scales. I understand that the sharp boundaries may suggest little cell communication, later in wing pattern development. However, the localization of clusters suggests cortex may impact communication between cells in early wing cell replication. This doesn't have much of an impact on their findings, but the authors may want to consider tempering the interpretation.*

We thank the reviewer for bringing this to our attention. We have changed our interpretation to incorporate the idea that communication between neighbouring cells may be contributing to the observed clustering of mutant cells (see lines 244-246).

*12) Ln 352-353. The authors state "…cortex is necessary for the development of lamellar structures in Heliconius scales.". This statement is too broad and encompassing. Figure 9 shows that cortex appears to be required to develop Type II and III scales with open upper lamina. In this regard, cortex is not required for the development of lamellar structures but rather required to regulate the development of architectural variation in lamellar structures. This may seem minor, but as worded it reads that "cortex is necessary for the development of lamellar structures", which I don't agree is supported by the data.*

We agree with the reviewer that our wording was incorrect. We have edited these lines to reflect the fact that *cortex* perturbations are accompanied by morphological changes. See lines 357-359.

*13) Ln 393-398. I find the authors attempt to explain the difference in the direction of cortex DGE between species to be quite brief and hand-waving. I'm not clear what is meant by "some relatively subtle developmental heterochrony between the two species would capture the state of differentially expressed genes in a different dynamic step." It seems the timing and levels of cortex expression could be quite precise and variable over short periods of time. In such cases, the limited sampling in each species could simply be detecting different points. In other words, it may be more informative for the reader to have the authors state that the current resolution may simply be insufficient to resolve and compare the precise cellular functions of cortex in the two species. And, that the difference between species suggests more detailed examination, perhaps even real-time expression data of cortex, may be needed.*

We agree and have changes the lines accordingly (see lines 400-405).

*14) Figure 10. I would consider including WntA in Figure 10. The timing of expression of WntA effecting melanic pattern is well resolved (5th instar), the authors discuss how WntA and cortex may interact, so it would seem reasonable to include WntA in Figure 10, so that the figure offered the best present model for the timing and placement of each major color patterning gene during wing development.*

We thank the reviewer for the suggestions. Upon design of the figure, we had previously considered including *WntA*. However, including *WntA* is tricky as *WntA* signaling does not determine strict colour fates, but rather shapes the spatial organization of the landscape. We thought about describing the landscape itself as a metaphor for the action of *WntA*, whereby it can shape the hills and throughs, delimiting the spatial arrangement of specific colour pattern elements. Or having the “scaffolding” underneath the landscape be represented as *WntA*, however, we believe this might be stretching the metaphor a bit and become confusing to the reader.

*15) Ln 510. Italicize H. melpomene and H. erato.*

This has been changed.

*16) Ln 684. Ray et al. 2019 not included in References.*

Apologies. This has now been added.

*17) Ln 885-886. McMillan et al. 2020 reference is incomplete.*

This has now been updated.

*18) Figure S15. Please remove "Lorem ipsum" from the "H. erato ridge periodicity" plot.*

Yikes. This has now been removed.

*19) The introduction lacks a short description of the concrete wing colour patterns of the studied Heliconius species. It would be helpful for some readers to have this information between lines 100-106: Colour stripes, location on the wing, colour polymorphism,…*

Thank you for the comment. We have added two sentences describing the phenotypes in more detail (See lines 106-111).

*20) I agree with a previous reviewer that the data on cortex and dome/wash differential gene expression could be removed from this manuscript (although it should be published elsewhere). However, in the current form the manuscript starts with results that are difficult to interpret and this is confusing. The rational to investigate cortex function is already given by Nadeau et al. 2016.*

We thank the reviewer for the comment. We have tried to shorten the section dealing with dome/wash by including most of the information in the supplementary. However, we agree with the reviewer who suggested in Comment #10 that it is necessary to include part of the information in order to justify running the analysis in the way that we did. We hope this is sufficient to improve the clarity of the manuscript.

*21) The manuscript could therefore gain clarity if the authors re-arrange their very interesting findings. 1) CRISPR mosaics to characterize cortex function and SEM analysis on scale type 2) Identification and characterization of the CRE's and the role in stripe determination 3) cortex expression analysis that does not seem to associate with a hindwing stripe.*

We thank the reviewer for the suggestion. In terms of the set up and rationale, we would like to keep the structure as is, if the editors and reviewers agree. We agree that Nadeau et al. (2016) show compelling evidence for the involvement of *cortex* in establishing pattern differences, however, their conclusions were drawn from microarray tiling experiments examining differences between *H. melpomene* morphs that did not differ specifically in the presence of a yellow bar (*H.m. plesseni* and *H.m. malletti*). We believed it necessary to expand on these analyses by showing *cortex* was also differentially expressed in association with the yellow bar phenotype, which is the phenotype of focus in our manuscript. With therefore think that starting with the DGE as the first experiment is important. We also think that, while it might improve clarity to leave out *dome/wash*, it would be difficult not to report that there is another locus among the 47 gene interval that shows promising patterns of differential gene expression.

*22) Mosaic loss of function of cortex apparently reveals a loss of melanic (Type II) and red (Type III) scales. The authors note sharp boundaries between wild-type and mKO scales and suggest cortex functions to be cell-autonomous with little or no communication between neighbouring cells. However, in Figure 4b and S10 they also report intermediate white/red scale phenotypes in scales of the "red band element". These scales sometimes have a patchy/granular red colouration. The authors further note an asymmetric deposition of pigment across the scale but should better point out what this means. Is the result conform with / contradicting their interpretation of a cortex mediated scale identity switch? Could the red pigment presence have non-autonomous sources from the surrounding tissue?*

We thank the reviewer for raising this important point. We have changed our interpretation also in line with Comment #11. Regarding the more continuous nature of the red scales, we have tried to elaborate this by including more information in the supplementary (See Comment #3). In terms of the interpretation for a direct Type III to Type I switch, we agree that this is not a simple as the case observed for Type II to Type I. However, we believe that the data still supports the idea that a *cortex* signal is required for “proper” development of Type III scales, as these display Type-I-like structure when in a *cortex* negative state.

*23) The Figures are very nice, but many legends lack a certain degree of precision in the description of each panel. The authors need to revise this with care, this has already been pointed out by a previous reviewer. Please indicate what the illustrations literally present. For example, Figure 1a, horizontal black bars are coding sequences? gene predictions? mapped transcripts? other elements? What reference genome assembly? Do they indicate the 47 candidate genes or just a small portion of it? Wing patterns are present on the dorsal or ventral side of the wing? Figure 1B: Are yellow and blue dots reported sampling locations?*

We apologise for the lack of clarity regarding the figure legends. The legends have been edited to include more detail, we hope this has improved clarity.

*24) Conclusion: The first part (lines 492 – 505) appears to be redundant with lines 511 -514 in the second paragraph. Maybe integrate the first paragraph into the discussion.*

We agree with the reviewer and have shortened the conclusion to remove redundant lines.